# Bulk Flow Optimisation of Amorphous Solid Dispersion Excipient Powders through Surface Modification

**DOI:** 10.3390/pharmaceutics15051447

**Published:** 2023-05-09

**Authors:** Danni Suhaidi, Yao-Da Dong, Paul Wynne, Karen P. Hapgood, David A. V. Morton

**Affiliations:** 1School of Engineering, Deakin University, Waurn Ponds, VIC 3216, Australia; danni.suhaidi@deakin.edu.au; 2Drug Delivery, Disposition and Dynamics, Monash Institute of Pharmaceutical Sciences, Monash University, Parkville, VIC 3052, Australia; 3Medicines Manufacturing Innovation Centre, Monash University, Clayton, VIC 3168, Australia; 4School of Engineering, Swinburne University, Hawthorn, VIC 3122, Australia

**Keywords:** amorphous solid dispersions, spray drying, L-leucine, bulk powder characterization, particle engineering, flowability enhancement, surface modification

## Abstract

Particulate amorphous solid dispersions (ASDs) have been recognised for their potential to enhance the performance of various solid dose forms, especially oral bioavailability and macromolecule stability. However, the inherent nature of spray-dried ASDs leads to their surface cohesion/adhesion, including hygroscopicity, which hinders their bulk flow and affects their utility and viability in terms of powder production, processing, and function. This study explores the effectiveness of L-leucine (L-leu) coprocessing in modifying the particle surface of ASD-forming materials. Various contrasting prototype coprocessed ASD excipients from both the food and pharmaceutical industries were examined for their effective coformulation with L-leu. The model/prototype materials included maltodextrin, polyvinylpyrrolidone (PVP K10 and K90), trehalose, gum arabic, and hydroxypropyl methylcellulose (HPMC E5LV and K100M). The spray-drying conditions were set such that the particle size difference was minimised, so that it did not play a substantial role in influencing powder cohesion. Scanning electron microscopy was used to evaluate the morphology of each formulation. A combination of previously reported morphological progression typical of L-leu surface modification and previously unreported physical characteristics was observed. The bulk characteristics of these powders were assessed using a powder rheometer to evaluate their flowability under confined and unconfined stresses, flow rate sensitivities, and compactability. The data showed a general improvement in maltodextrin, PVP K10, trehalose and gum arabic flowability measures as L-leu concentrations increased. In contrast, PVP K90 and HPMC formulations experienced unique challenges that provided insight into the mechanistic behaviour of L-leu. Therefore, this study recommends further investigations into the interplay between L-leu and the physico-chemical properties of coformulated excipients in future amorphous powder design. This also revealed the need to enhance bulk characterisation tools to unpack the multifactorial impact of L-leu surface modification.

## 1. Introduction

In recent years, there has been substantial interest in the pharmaceutical particle engineering research community for optimising spray-dried formulation powders and conditions to synthesise smaller-sized (≤10 µm) amorphous solid dispersions (ASDs) [1,2]. Typically, ASD formulations are used to improve the stability of biomacromolecules [3] and dissolution characteristics of poorly water-soluble active pharmaceutical ingredients (APIs) [4,5]. Because of their small particle size and amorphous physical nature, these spray-dried ASD powders have both a high surface-area-to-volume ratio and high surface energy, which increases the impact of cohesive interparticular forces, such as van der Waals interactions, on bulk behaviours [6]. Most industrially relevant ASD powders are postprocessed in downstream operations consisting of feeding, blending, filling, and tableting [7,8]. 

This poses a challenge, as formulated highly cohesive fine ASD powders tend to agglomerate and suffer from low bulk powder flowability. These ramifications include transport issues such as bridging or arching, which results in decreased operational efficiency and reduced yield [9]. Furthermore, benefits in the increased surface area for enhanced dissolution can be lost due to stronger agglomeration. Relatively few publications have explored potential solutions to this issue [10]. 

One promising avenue is the use of a coprocessing approach to modify the surface characteristics of the ASD powder materials. Zhou et al. demonstrated that the surface coating of fine lactose powders with magnesium stearate using mechanofusion successfully improved powder bulk flowability [11]. However, magnesium stearate has been reported to reduce powder dissolution rates, rendering it a less appealing option [12]. An alternative is the introduction of coexcipient materials during the spray-drying particle formulation process to achieve surface modification. Previous studies have explored the utilisation of amino acids, including arginine, aspartic acid, L-leucine (L-leu), l-isoleucine, and phenylalanine, to enhance pulmonary performance [13]. Among them, L-leu has been consistently found to have a substantial effect on improving the aerosolisation potential, attributed to its surface activity [14].

L-leu is an aliphatic α-amino acid with a hydrophobic isobutyl side chain frequently used in inhalation powder formulation research to improve powder dispersibility for pulmonary delivery [15]. In research studies in this area, it was reported and proposed that during the early stages of drying, the relative L-leu droplet surface concentration increased until it reached a supersaturation point, where it underwent a self-assembly process to form the observed partially ordered crystalline layers [16,17]. This shell interrupts the mass transfer of water vapour to the external environment, consequently creating a distinct corrugated morphology [18]. The effectiveness of L-leu was theorised to be related to its observed surface crystallinity, and its initial formulation concentration would be a deciding factor in influencing bulk flowability resulting from surface impact [16]. 

Published state-of-the-art research reports coformulating L-leu with APIs to leverage its properties as a potential dispersion, stability, and dissolution enhancer [4,5,15,19,20,21]. Mangal et al. highlighted the potential of developing a universal multifunction tablet excipient platform when their research showed the dramatic impact of different concentrations of L-leu on the flowability of polyvinylpyrrolidone K10 (PVP K10)-based formulations [22]. As a follow-up, this study aimed to explore the potential powder flow enhancement of a coformulation of L-leu with standard amorphous-forming excipients ubiquitously used in the pharmaceutical industry. Therefore, this study addresses this research gap by studying and comparing candidate excipients with differing physico-chemical properties to determine bulk powder outcomes from cospray drying with or without L-leu. The intention was to develop knowledge to underpin potential multifunctional particle-engineered delivery platforms suitable for the ASD powder formulation needs of future APIs. 

Mannitol is one of the most used sugar-based excipients in commercial products; consequently, there has been a glut of scientific research exploring its physico-chemical properties and compatibility with L-leu [23,24]. However, mannitol in this form rapidly recrystallises from its initial amorphous nature, making it less attractive for ASD powder formation. Thus, this study will prototype various alternative coprocessed prototypes using emerging excipients from both the food and pharmaceutical industries with greater potential to remain amorphous and then examine the resulting bulk powder behaviours on coformulation with L-leu. 

Maltodextrin was selected as a polysaccharide consisting of D-glucose subunits typically used as a binding agent in tableting and carrier during spray drying [25]. Maltodextrin has been successfully formulated as an amorphous carrier to improve the bioavailability of various nutraceutical products [26,27]. Trehalose is a disaccharide nonreducing sugar with a relatively high glass transition temperature and has been proposed as a strong candidate to remain in a metastable amorphous form. Currently, it is coformulated in several therapeutics such as: Herceptin^®^, Avastin^®^, Lucentis^®^, and Advate^®^ as a protein-stabilising agent [28]. Recently, it has gained academic interest because of its potential as a spray-drying carrier for the effective pulmonary delivery of next-generation APIs [29,30,31]. Gum arabic is a heterogeneous collection of polysaccharides and glycoproteins with applications as an emulsifying agent, stabilising agent, and tablet binder [32]. Additional interest stems from its reported physico-chemical benefits as an antioxidant and its role in lipid metabolism [33]. 

PVP, a water-soluble polymer composed of N-vinylpyrrolidone subunits, is a promising amorphous solid excipient. PVP is a nontoxic and stable excipient formulated into a large range of novel controlled delivery systems [34]. PVPs are considered desirable excipients in solid dispersions because of their mucoadhesive properties that prolong their retention time in mucosal membranes [35,36]. Published studies have already shown the effectiveness of L-leu surface modification in improving PVP K10 bulk flowability. To extend our understanding, this study investigated the use of PVP K10 (10,000 Da) and the larger molecular weight PVP K90 (1,300,000 Da), which differ in their viscosity forming characteristics. These PVPs serve as benchmarks from past work to evaluate the flowability changes of other excipient powders. 

The last excipients explored were two grades of soluble methylcellulose ether hydroxypropyl methylcellulose (HPMC), which is commonly used as a tablet binder or in amorphous matrices for extended release [37,38]. The two HPMC grades investigated in this study were E5LV and K100M. They differ in their chemical structures because of the different functional groups substituted in their structures. A prominent difference between the two HPMCs is their viscosity. HPMC E5LV has a lower viscosity of 6 mPa·s than HPMC K100M at 100,000 mPa·s. This difference in viscosity can be attributed to the difference in their substitution levels as well as the size and weight of their respective molecules [39].

Currently, conventional approaches for characterising potential ASDs involve understanding their behaviour at the molecular level using thermal and spectroscopic techniques and evaluating their pharmacokinetic behaviour through in vitro and in vivo studies [40,41]. These may provide some insight into the mechanism underlying the coprocessing behaviour with, for example, L-leu, but these techniques do not inform research on the holistic bulk behaviours of these powders [17,22]. Therefore, this study evaluates each powder composition formulated with or without L-leu using a range of standard and modified bulk powder tests from the Freeman FT4 powder rheometer, which were selected to gain an understanding of the effect of L-leu on the resulting bulk powder cohesion properties. In addition, scanning electron microscopy (SEM) and particle sizing were used to further differentiate the formulations for a better perception of structure–performance properties from coprocessing, and powder X-ray diffraction (XRD) was employed to examine the crystalline-amorphous nature.

## 2. Materials and Methods

### 2.1. Materials

Maltodextrin DE18 and gum arabic were acquired from the Melbourne Food Depot (Melbourne, Australia). L-leu was purchased from Sigma-Aldrich (Castle Hill, Australia), and trehalose Powder was sourced from Swanson (Melbourne, Australia). Polyvinylpyrrolidone K10 (Average molecular weight 10,000 Da) and K90 (Average molecular weight 90,000 Da) were sourced from Sigma-Aldrich (Castle Hill, Australia). Hydroxypropyl methylcellulose E5LV (Methocel E5 Premium LV) was purchased from DuPont chemicals (Mississauga, ON, Canada), and hydroxypropyl methylcellulose K100M was purchased from Sigma-Aldrich (Castle Hill, Australia).

### 2.2. Formulation and Spray Drying

Table 1 lists the formulation composition of each excipient. These were dissolved in aqueous solutions in accordance with the desired L-leu concentrations (0, 2.5, 5, 7.5, 10, 15, 20, 25, and 30 wt% dry basis). The mass of each component was weighed using a mass balance to within ±5% of the ideal target mass, and subsequently mixed with 400 mL of demineralized water (≤10 ppm total dissolved solids) using a magnetic stirrer (800 rpm, 35 °C). Spray drying was conducted using a Büchi B290 mini benchtop spray dryer (Büchi Laboratory Equipment, Flawil, Switzerland) with a standard 0.5 mm two-fluid nozzle. The standard operating conditions were as follows: T_inlet_, 125 °C; aspirator rate, 35 m^3^/h; feedstock flow rate, 7.5 mL/min; and T_outlet_, 76 ± 2 °C. After spray drying, the powders were quickly collected in a sealed container and stored away from direct sunlight to minimise environmental exposure. 

### 2.3. Scanning Electron Microscopy

The surface morphologies of the formulations were imaged using a JSM-IT300 microscope (JEOL, Tokyo, Japan). A small amount of powder was fixed onto carbon tape on an aluminium stub, and the excess powder was removed using compressed air. It was subsequently given a 5 nm platinum coating using an ACE600 sputter coater (Leica, Wetzlar, Germany). The platinum-coated stubs were loaded into the SEM and imaged using an electron voltage of 10 kV at a working distance of 12 mm. 

### 2.4. Particle Size Analysis

The particle size distribution data were acquired using a Camsizer X2 with an X-Dry Module and X-Jet cartridge (Retch Technology GmbH, Haan, Germany). Principally, Camsizer X2 functions based on ISO 1332202, where two cameras (a zoom for smaller particles and a base for larger particles) worked in conjunction with digital image analysis to derive particle size and shape distributions. Briefly, the powder was fed into the system using a hopper mechanism, where it encountered a compressed air stream that dispersed the powder into individual particles flying perpendicular to the sightlines of both cameras. These particles were illuminated using a light source, and the generated shadows were imaged using two cameras. Spray-dried powders generally adopt a spherical morphology; therefore, image analysis was conducted using the Camsizer X2′s X_area_ protocol. The X_area_ protocol determined the particle diameter by calculating the diameter of an equivalent circle with the volume of a sphere of diameter X_area_ (Equation (1)).
(1)$$X_{\mathrm{area}}=\sqrt{\frac{4A}{\pi }}$$

Each formulation also had velocity adaptation performed to eliminate the risk of overlapping particle measurements. Three measurements were taken for each formulation and the average value was calculated to obtain the desired volume-based D_10_, D_50_, D_90,_ and span. 

### 2.5. Powder X-ray Diffraction

Powder X-ray diffraction (P-XRD) analysis was performed using a Shimadzu 7000 L X-Ray Powder Diffractometer (Shimadzu, Kyoto, Japan). Scanning was performed from 5° to 35° at 2θ. The crystalline status of several formulations was evaluated using diffraction patterns. 

### 2.6. Powder Bulk Flowability Characterization

Powder bulk flowability was characterised using a Freeman FT4 powder rheometer (Freeman Technology, Worcestershire, UK). This well-established instrument quantifies powder flowability through accessories, such as blades and shear heads, which are axially inserted into a powder bed while simultaneously rotating them [42]. For analytical purposes, four separate test protocols available for FT4 were used to quantify each spray-dried formulation: shear cell testing, stability and variable flow rate testing, modified stability testing, and permeability testing.

#### 2.6.1. Shear Cell Testing

The shear cell test is based on the principle of applying both vertical and rotational stresses using a shear head on a sample powder bed [43]. Initially, an appropriate volume of each formulation was loaded into a standard 25 × 10 mL split-ring vessel, where a vented piston was used to precondition the powder bed to 9 kPa. Subsequently, the shear head was lowered into the powder bed until the predesignated normal stresses of 3, 4, 5, 6, and 7 kPa were reached. Slow rotation was then applied to the powder bed at a set rate until the powder bed experienced incipient flow. The maximum shear stress observed prior to the incipient flow was then recorded for each normal stress examined, and the data were plotted to generate a yield locus (Equation (2)).
(2)$$\tau =\sigma \tan\left(\eta \right)+C$$
where τ denotes the shear stress, σ is the normal stress, η is the angle of internal friction (AIF), and C is the cohesion force. For this analysis, the flow function coefficient (ffc), AIF and Cohesion parameters were used to compare each formulation. The cohesion parameter was derived by extrapolating the yield loci to identify the y-axis shear-stress intercept, which represents the strength of a powder under zero confining stress [6]. The ffc represents the ratio between the major principal stress (MPS) and unconfined yield strength (UYS). The AIF was defined as the angle of the line drawn from the origin of the shear-stress vs. normal-stress graph towards the preshear data point.

#### 2.6.2. Stability and Variable Flow Rate Testing

The stability test was based on the principle of axially inserting a blade into a powder bed while simultaneously rotating it at a set rate and angle. Each formulation was loaded into a 25 × 25 mL split-ring vessel and underwent two conditioning cycles prior to testing. The conditioning was intended to return the powder bed to a similar initial compaction stage to minimise errors between runs of the same sample. The protocol dictated that each test comprised 7 identical consecutive test cycles, in which the blade operated at 100 mm/s. Subsequent cycles were referred to as the variable flow rate test, where the blade speed was varied as 100 mm/s, 70 mm/s, 40 mm/s, and 10 mm/s. 

Several data metrics were derived from this protocol: the basic flow energy (BFE), specific energy (SE), and flow rate index (FRI). The BFE represents the total aggregated energy required for the blade to progress through the powder bed during its downward motion. This was calculated using the seventh downwards test cycle of the protocol. SE measures the total energy needed to move the blade in an upward direction out of the powder bed; compared to BFE, it does not account for the force exerted to compress the powder bed during downwards motion. SE was calculated using the average value of the total energy required during the upward traverse of conditioning cycles six and seven then divided by mass. Finally, the FRI is a measure of powder bed sensitivity to changes in shear rate, defined as the ratio between the total energy recorded at a blade speed of 10 mm/s vs. 100 mm/s. 

#### 2.6.3. Modified Stability Testing

FRI data from the previously mentioned flow rate test highlighted the poor differentiating behaviour between the formulations. A previous review indicated that different formulations experience different shear sensitivities. Therefore, the standard FT4 stability protocol was modified in consultation with Freeman Technology staff to explore this phenomenon. The standard test was altered from using 100 mm/s to 60 mm/s. It should be noted that procedures for data collection, such as BFE and SE, were not altered to ensure that comparison between datasets was possible. 

#### 2.6.4. Permeability Testing

The permeability testing comprised three initial conditioning steps to ensure identical initial compaction states in the powder beds. Subsequently, a vented piston was used to bring the powder bed to specific normal stresses (1, 2, 4, 6, 8, 10, 12, and 15 kPa), and the pressure drop incurred from a specific air velocity was measured. The formulations were loaded into a 25 × 10 mL split-ring vessel for testing. The air supply module was cleaned and calibrated between every formulation to minimise the impact of powder accumulation. Experiments were conducted in triplicate for each formulation, and the reported results are the averaged values. 

#### 2.6.5. Statistical Analysis

Statistical analyses were performed using Microsoft Excel (Microsoft Corporation, Redmond, WA, USA). Changes in bulk flowability of the different excipient powder coformulations with and without L-leu were evaluated using a one-way ANOVA test with *p* < 0.05, regarded as significant.

## 3. Results

### 3.1. Scanning Electron Microscopy

Selected representative electron microscopy images of the formulations are shown in Figure 1, Figure 2, Figure 3, Figure 4, Figure 5 and Figure 6. Morphologically, the maltodextrin/L-leu formulations displayed the most visually observable changes with increasing leucine concentrations (Figure 1). At 0 wt% L-leu, a substantial number of the spray dried powders appeared as typical collapsed spheres which could be described as ‘blood cells’. The powders physically presented as more agglomerated than those containing L-leu. 

At L-leu concentrations of 5 wt%, some unusual structures appeared, notably a unique ‘cupcake’ morphology not seen in other formulations (Figure 1B). We propose that this resulted from L-leu influencing the mechanical properties of the surface film of the drying droplet. As the internal vapour pressure increased, a localised puncture appears to have occurred, which likely led to an irregular structure. Starting from a 10 wt% concentration, the powders began to exhibit a more heavily indented and wrinkled morphology, typically associated with L-leu surface-modified powders. However, above 20 wt%, the wrinkled morphology was accompanied by more spherical structures with apparent surface flaking. 

PVP K10/L-leu formulations displayed degrees of morphological changes with increasing L-leu, as previously reported (Figure 2). At 0 wt%, the powder exhibited a smooth-dimpled morphology, which progressed to a more corrugated surface at 5 wt%. At concentrations of ≥10 wt%, ‘collapsed’ morphologies were observed. As previously theorised, above a certain concentration threshold, L-leu was able to form an outer shell with sufficient mechanical and transport resistance to entrap the escaping water vapour. This results in an increased internal vapour pressure, leading to expansion and eventual structural failure, resulting in the observed collapsed morphology [22].

Unlike PVP K10, PVP K90 demonstrated previously unreported complications with L-leu during spray drying (Figure 3). At concentrations of 2.5–10 wt%, spray drying generated macroscale fibrous byproducts at the spray nozzle which severely affected the yield (Figure 3B,C). In each case, long fibrous spindles of material reminiscent of silk fibres formed within the drying chamber, preventing the recovery of most free powder. Consequently, no powder was recovered at 2.5 wt% and 7.5 wt% L-leu concentrations, preventing any bulk powder characterisation of these concentrations. However, no fibrous byproducts were generated at concentrations of ≥15 wt%, with the collected powders sharing corrugated morphologies similar to those of the PVP K10 formulations. Notably, these fibrous formations did not occur during the initial feedstock formulation or in the pure PVP K90 spray-drying runs. 

Trehalose-based powders displayed a different morphological progression from the other powders (Figure 4). SEM of individual particles did not display an apparent transition from smooth dimpled spheres to corrugated structures. Instead, the powders maintained a mostly lightly dimpled spherical shape from 0 wt% to 30 wt%. Gum arabic is a heterogeneous material that is composed of polysaccharides and glycoproteins. SEM images showed that gum arabic exhibited a gradual morphological change as L-leu concentration increased (Figure 5). At 0 wt%, it exhibited similar ‘blood-cell’ like particles, like pure maltodextrin powder. Increasing the L-leu concentration resulted in the same wrinkled corrugated morphologies associated with L-leu. However, unlike homogeneous excipients, gum arabic required a higher L-leu concentration to confer the same morphological shifts.

Finally, electron microscopy showed that there were no discernible morphological differences between spray-dried pure HPMC and those coprocessed with L-leu. At all L-leu concentrations, both the HPMC E5LV and K100M (K100M in Appendix A as Figure A1) powders presented themselves as smooth-dimpled powders (Figure 6). 

### 3.2. Particle Size and Size Distribution

The dispersion pressure was optimised through a pressure-titration process, which showed that 100 kPa provided sufficient force to deagglomerate each formulation. All formulations displayed a narrow monomodal particle size distribution; the particle size data are summarised in Table 2. Most spray-dried powders had a relatively small particle size distribution (D_50_: ~4–6 µm), whereas known viscosity-modifying excipients, such as PVP K90, HPMC E5LV, and K100M, had a relatively larger particle size distribution (D_50_: ~8–10 µm). Formulating these excipients in the same volume of solvent would result in higher feedstock viscosity. This is believed to impact the atomisation efficiency in the spray nozzle, leading to larger droplet sizes, which increase the mean particle size of the dried powders [44]. There are conflicting reports on the effect on the particle size distribution. Previous reports by Mangal et al. showed that higher L-leu concentrations resulted in larger particle size distributions [22]. However, Ferdynand et al. observed that L-leu decreased the particle size distribution owing to its surfactant properties [45]. This was observed in both HPMC E5LV and K100M, where the particle sizes decreased slightly with more coprocessed L-leu. Furthermore, the particle size is known to play an integral role in powder cohesion. However, the particle size data for nonviscosity influencing excipients showed no substantial difference; therefore, size should play no substantial role in influencing interparticular interactions. Hence, this study can directly attribute any bulk flowability improvement to L-leu surface modification. 

### 3.3. Powder X-ray Diffraction

The P-XRD diffractogram analysis of several formulations was conducted to evaluate the solid-state characteristics of L-leu (Figure 7). Figure 7I shows the diffractogram of pure unspray-dried L-leu powder, which was apparent based on the existence of several distinctive peaks of its crystalline nature. The spray-dried powders that were not coprocessed with L-leu did not contain any distinct diffraction peaks, indicating that the powder existed in an amorphous form (Figure 7A,C,E,G). A diffuse peak was noted at 10° for all excipients. We theorise that this diffuse peak likely represents the aggregate chain length of polymeric excipients. This broadening effect was attributed to the random unordered conformations adopted by each individual polymeric chain [46]. Conversely, at a L-leu concentration of 20 wt%, all formulations displayed distinct 6° and 19° 2θ diffraction peaks along with a diffuse background (Figure 7B,D,F,H). Compared to the profile in Figure 7I, this is indicative of an amorphous core surrounded by a partially ordered L-leu crystalline shell. Sou et al. theorised that L-leu arranges itself into a strong two-dimensional lamellar form, but it failed to fully achieve a three-dimensional order [17]. Unlike other excipients, the HPMCs exhibited a degree of peak broadening in their diffractograms (Figure 7H). We theorise that this was a consequence of the physico-chemical properties of HPMC interfering with L-leu’s ability to self-assemble into a coherent crystalline structure.

### 3.4. Shear Cell 

The bulk powder flowability of each formulation under confined stress conditions was evaluated using an FT4 powder rheometer shear cell test, and the results are summarised in Figure 8. The examined data points were cohesion, flow function coefficient (ffc), and AIF. 

#### 3.4.1. Cohesion Data

Cohesion data displayed a differentiation of each powder as L-leu content increased, with two distinct behaviour patterns observed (Figure 8A). Most pure spray-dried excipients displayed a higher degree of cohesiveness, which was expected for micron-sized fine powders (cohesion values ≥2 kPa). The exceptions were the HPMC formulations (E5LV and K100M), which exhibited a relatively low degree of cohesion (≤1.5 kPa). For both HPMCs, increasing L-leu concentration did not result in improved flowability. Instead, it slightly increased cohesion values at concentrations ≥15 wt%; for all other excipient powders, cohesion values decreased more substantially, indicating flowability improvements directly tied to increases in L-leu concentration. 

Both maltodextrin and trehalose formulations achieved the most substantial flowability improvements at L-leu concentrations as low as 2.5 wt% (*p* < 0.05). These cohesion improvements plateaued at ≥5 wt% (*p* < 0.05) with maltodextrin (0.74 ± 0.26 kPa and trehalose (0.61 ± 0.27 kPa respectively). Further increases in concentration did not confer additional benefits, and in the case of trehalose, concentrations ≥10 wt% appeared to gradually decrease its flowability performance from 0.62 ± 0.053 kPa at 10 wt% L-leu to 0.89 ± 0.04 kPa at 30 wt% L-leu (*p* < 0.05). Additionally, increasing L-leu concentrations led to flowability improvements in PVP K10 formulations, reaching a maximum reduction of cohesion at 7.5 wt% and reaching a cohesion of 0.56 ± 0.025 kPa (*p* < 0.05). Consistent with the conclusions reported by Mangal et al., increasing the concentration above 10 wt% did not result in any further substantial improvement in bulk powder flowability [22]. Although the required concentration of L-leu was relatively higher, the maximum cohesion reduction for PVP K10 formulations marginally exceeded that of maltodextrin and trehalose powders (*p* < 0.05). Formulations of both PVP K90 and gum arabic required a greater level of L-leu to achieve similar cohesion value improvements. At L-leu concentrations of ≥20 wt%, flowability improvements were comparable to those of other excipients such as maltodextrin, trehalose, and PVP K10 (*p* < 0.05).

#### 3.4.2. Flow Function Coefficient (ffc)

ffc has been considered a benchmark measure of powder flowability in the materials industry [47], and the most common interpretation of this value is shown in Table 3. In this study, only the preshear consolidation stress of 9 kPa was investigated. This largely followed the same bulk property trend as the cohesion data; in general, an increase in L-leu concentrations resulted in an increase in flowability (Figure 8B). 

The relationship between cohesion and ffc values was previously reported by Wang et al., who showed that these parameters derived from low-cohesion powders with the same preconsolidation stress had a statistically substantial inverse correlation with low-cohesion powders [48]. The results of this study further support this relationship.

HPMC formulations could be classified under the ‘cohesive’ category, as the majority of their ffc values remained below four. For maltodextrin, trehalose, and PVP K10, a L-leu concentration of ≥5 wt% could be classified as easily flowing. Gum arabic only became ‘easy-flowing’ after a concentration of 15 wt%. No powders could be definitively identified as ‘free-flowing” owing to the high degree of variability in the recorded data between the different runs.

#### 3.4.3. Angle of Internal Friction (AIF)

AIF was derived to represent the bulk friction of a powder bed during incipient flow. It was determined by drawing a linear line of the best fit from the origin of a shear-stress vs. normal-stress graph towards the preshear data point. Contrary to both cohesion and ffc, AIF did not clearly differentiate between formulations (Figure 8C). Increasing the L-leu content resulted in only a slight reduction in the AIF of maltodextrin and trehalose at ≥20 wt%. No substantial changes were noted for gum arabic- and HPMC-based formulations. Instead, L-leu had a pronounced impact on the AIF of the PVP K10 formulations. Initially, the AIF increased from 0 wt% to 2.5 wt% (25.34° ± 0.18, 0 wt% vs. 31.08° ± 1.21, 2.5%) and then decreased to a value substantially below its starting point (25.34° ± 0.18, 0 wt% vs. 20.11° ± 1.37, 30 wt%) (*p* < 0.05). In the case of PVP K90, at L-leu concentrations of 5 wt% and 10 wt%, the fibrous byproduct appeared to have contributed substantially to the frictional resistance of the powder bed, greatly increasing the AIF (25.44° ± 0.39, 0 wt% vs. 41.79° ± 0.89, 10 wt%) (*p* < 0.05). This was followed by a significant reduction at concentrations of ≥15 wt%, reaching an AIF of ~21° (*p* < 0.05).

### 3.5. Stability and Variable Flow Rate

#### 3.5.1. Basic Flow Energy (BFE)

The BFE data derived for each formulation using the FT4 standard stability test protocol are shown in Figure 9A. The results showed that increasing the L-leu content led to a general decrease in the BFE. The BFE itself was defined as the amount of total energy required by the blade to move through the powder bed in a downward anticlockwise motion. Therefore, the BFE data were effectively the inverse of the bulk flowability. Thus, when the BFE values decreased, the flowability of the powder increased. Unlike the previous shear cell data, the maltodextrin-based formulations experienced a more gradual decrease in BFE. A L-leu concentration of ≥10 wt% was used to achieve the maximum BFE reduction (≤20 mJ, *p* < 0.05). Conversely, PVP K10 required a comparatively lower concentration of L-leu (≥5 wt%) to achieve the same BFE reduction (≤20 mJ, *p* < 0.05). Trehalose powder exhibited the most substantial reduction in BFE, reaching a maximum reduction (≤20 mJ, *p* < 0.05) at concentrations as low as ≥2.5 wt%. Gum arabic formulations required a high amount of coprocessed L-leu (≥20 wt%) to achieve a BFE of ≤20 mJ (*p* < 0.05). Interestingly, the BFE was able to differentiate the fibrous byproduct found in PVP K90 powders at L-leu concentrations of 2.5–10 wt%, where BFE values increased to 134 ± 60 mJ at 5 wt% from 53 ± 4 mJ of pure spray-dried PVP K90. Initially, it was presumed that the BFE could have provided insight into the formulation behaviour at a lower-stress regime. However, it became apparent that the BFE provided insufficient differentiation between each excipient once sufficiently surface-modified with L-leu. Regardless of excipient or additional L-leu coprocessing, BFE was not able to detect any further improvements below the 20 mJ threshold. 

#### 3.5.2. Specific Energy (SE)

In contrast to the BFE, the SE measured the upward clockwise motion of the blade as it left the powder bed. In effect, it provided an evaluation of the flow characteristics in unconfined and uncompacted states, unlike the shear test and downwards motion-based BFE value. Therefore, it eliminated the energy component required to compact the bed and represented a different powder characteristic. In addition, it was also normalised for the mass of the powder bed, minimising the impact of differences in the material bulk density on the total energy measured. As a general rule, a low SE value is associated with low material cohesion, with an SE value <5 classified as low cohesion. 

As shown in Figure 9C, contrary to cohesion data from the shear cell test, under unconfined flow, maltodextrin-based formulations required a noticeably higher concentration of L-leu (≥10 wt% vs. ≥5 wt%) to achieve low cohesion (3.53 ± 0.45 mJ/g, *p* < 0.05). In contrast, PVP K10 formulations reached low cohesion (3.79 ± 0.16 mJ/g) at a lower L-leu concentration compared to unconfined flow (≥5 wt% vs. ≥ 10 wt%). The threshold for maximum flowability improvement through L-leu coprocessing remained the same for trehalose and gum arabic powders, requiring concentrations of ≥5 wt% and ≥20 wt%, respectively. These results highlight the differences in the behaviour of these surface-modified powders under confined and unconfined flow conditions. 

#### 3.5.3. Flow Rate Index (FRI)

FRI is a flow property used to characterise bulk powder sensitivity to variations in the flow rate. Generally, cohesive powders are more sensitive to flow rate changes, with an FRI value of >3 associated with high cohesion. Figure 10A shows that coprocessing maltodextrin and trehalose powders with low concentrations of L-leu dramatically increased their FRI values (FRI≥3 at 5 wt% and 2.5 wt%, respectively). However, the FRI values gradually decreased as the L-leu concentration increased, eventually dipping back to below 3 at 30 wt%. This pattern is apparent in Figure 10B,C, where increasing L-leu in maltodextrin and trehalose powders reduced the gradient of these graphs, signifying a decreasing sensitivity towards flow rate changes. 

L-leu coprocessing mostly did not adversely affect the flowrate sensitivity of the PVP K10 and PVP K90 powders. Figure 10A shows that, at 5 wt%, the PVP K10 powders experienced a substantial increase in the FRI values; however, this value plateaued at its original level at ≥10 wt%. Furthermore, Figure 10C highlights PVP K10s with a generally low flowrate sensitivity at most L-leu concentrations. A pattern repeated with PVP K90 powders, regardless of L-leu concentration, flowrate sensitivity, and FRI values, remained consistent (Figure 10F). Conversely, gum arabic powder became increasingly more sensitive to the flowrate as the L-leu concentration increased, reaching a high of 3.82 ± 0.27 at 20 wt% (Figure 10A,F). 

It was clear that the flowrate sensitivity was a bulk powder characteristic affected by L-leu coprocessing. Further investigations are required to better understand the mechanistic underpinnings of this phenomenon. This was coupled with another noteworthy behaviour: during the stability test, formulations coprocessed with L-leu required several test cycles before they reached a steady state (Figure 9E). We theorise that this was a consequence of the powders needing to reach a steady state of compaction when exposed to downward motion of the blade. Because the FRI value only took one total energy measurement at every blade speed, it was considered that these powders did not reach a steady state. Therefore, in response to these two observed powder behaviour issues, a modified stability test was conducted to determine whether it could provide a better powder characterisation. 

### 3.6. Modified Stability 

Here, the standard stability test was modified to run at a lower blade speed; instead of seven cycles operating at 100 mm/s, they would operate at 60 mm/s. Maltodextrin powder was used to evaluate the effectiveness of the modified approach. A comparison of Figure 9E,F shows that this blade speed provides better powder differentiation. Figure 9E shows that the original higher flowrate did not distinguish between maltodextrin coprocessed with 10, 20, and 30 wt% L-leu. The same test, operating at lower speeds, was able to distinguish between these concentrations (Figure 9F). In addition, it was confirmed that these formulations still required several test cycles before they provided a steady-state total energy measurement. Altering the flowrates also led to increased differentiation in the extracted BFE and SE values. Trehalose exhibited the most substantial reduction in BFE, reaching a maximum reduction (≤20 mJ) at concentrations as low as ≥2.5 wt%. However, at a slower flow rate, the trehalose powders had higher BFEs than the maltodextrin and PVP K10 formulations (Figure 9A,B). A similar result was observed for SE data. The specific energy was derived from the upward motion of the blade during conditioning cycles six and seven, there were no alterations in its flow rate between the original and modified tests. A comparison of Figure 9C,D shows the differences between the measured values. We theorised that this was because the different flowrates during the test cycles led to different consolidation states of the powder, thereby indirectly influencing the specific energy measurement.

### 3.7. Permeability 

Permeability testing combines both compaction and air entrapment to characterise each formulation. It provided a relative assessment of the packing efficiency of each powder under different normal-stress regimes, as packing controls the bed porosity and thus the resistance to airflow. Figure 11 summarises the permeability test results for several excipients of interest. It shows two types of bulk behaviour profiles. The first, where the packing efficiency improved from L-leu coprocessing, was more noticeable under higher normal-stress regimes. PVP K10, trehalose, and HPMC were classified under this category. Except for 0 wt% and 2.5 wt%, PVP K10 formulations experienced a narrow range of pressure drop across the bed (8.6 ± 1.7 mBar, 25 wt% to 12.3 ± 2.1 mBar, 10 wt%). However, as the normal pressure increased, substantial differentiation was observed (Figure 11B). A similar pattern was also observed with the trehalose and HPMC formulations; at low normal pressures (1 kPa), the powders had a narrow range of pressure drops between the different L-leu concentrations. At higher normal pressures (15 kPa), the recorded pressure drops of each trehalose formulation were distinct from one another (Figure 11C,E). Although relatively modest in its improvement of packing efficiency, data from Figure 11E suggest that L-leu coprocessing had a positive impact on the performance of HPMC powders. 

The second category of powders had great differentiation between each L-leu concentration, regardless of the applied normal stress. Both maltodextrin and gum arabic powders displayed this behaviour (Figure 11A,D). At both low and high normal pressures, increasing the L-leu concentration generally resulted in a higher pressure drop across the powder bed. From these data, it could be inferred that the larger the increase in pressure drop as the L-leu concentration increased, the greater the improvements conferred towards packing efficiency. Therefore, PVP K10 and gum arabic benefitted the most from L-leu coprocessing. 

## 4. Discussion

The recent focus of academic research on L-leu in the coformulation of powders has been on understanding the mechanistic reasons behind its surface deposition and the ongoing self-assembly and crystallisation processes. However, studies have not evaluated its comparative effects when coprocessed into powders with different amorphous-forming excipient groups, especially its effect on powder flow. Therefore, the primary goal of this study was to investigate the effect of L-leu on the bulk flowability of excipients coprocessed at varying levels. 

Qualitative observation of the SEM images compiled during this study showed that L-leu surface modification resulted in a wide range of different morphologies based on the level of the coformulated excipient and excipient nature. Many morphologies have been previously observed, while some of our observed morphologies relating to L-leu levels have not been previously reported. The deleterious impact of a low concentration (2.5–10 wt%) L-leu on PVP K90, forming fibrous spindle byproducts, has not been reported. In addition, the formation of flaky spherical structures in maltodextrin does not conform to the previously reported patterns of a more corrugated or collapsed morphology.

Another example of morphological differences is the comparison between PVP K10 and the trehalose-based formulations. The PVP K10 particles exhibited increased corrugation up to ≥10 wt% L-leu, where the collapsed morphologies became prominent. In contrast, the trehalose powder retained a corrugated dimpled spherical morphology, regardless of L-leu content. The morphological progression of both excipients had been examined in past publications [22,49]. Vehring et al. theorised that spray-dried powder morphology could be predicted using the Péclet number [18]. The Péclet number is defined as the ratio between the droplet surface evaporation rate and the diffusivity of the excipient material within the droplet. Given that spray-drying conditions remained constant for all formulations, the only differences were the excipient physico-chemical properties and L-leu concentrations. PVP K10 has a larger molecular structure than trehalose and is a known film-forming material [50]. Because molecular size affects the diffusivity of a material, PVP-based formulations would have a higher Péclet number than trehalose. Therefore, we theorise that a combination of PVP K10 surface accumulation and a coherent L-leu crystalline shell that forms at ≥10 wt% concentration offers stronger resistance, which restricts the ability of water vapour to escape from the droplet core, leading to collapsed morphologies. In contrast, trehalose is not classified as a film-forming agent. Hence, trehalose should diffuse more readily within the drying droplet, resulting in the formation of more spherical particles. 

Increasing the L-leu content substantially improved the bulk flowability performance of the excipients tested, with the notable exception of HPMCs. Two mechanisms of action have historically been used to account for flowability improvements: (a) surface corrugation, which results in decreased particle contact areas, and (b) a coherent L-leu shell, which decreases the surface energy of spray-dried formulations. These mechanisms are not mutually exclusive and can work in complement, but our results presented here indicated that improvements across the different excipients were not simply explained by observed surface corrugation, and that a coherent L-leu shell is a more appropriate explanation for flowability improvements. For example, consider the case of trehalose and HPMC. Increasing the concentration of L-leu did not result in observed morphological changes. However, bulk characterisation data showed that coprocessing improved trehalose flow characteristics. In contrast, HPMC powders did not exhibit any substantial bulk flowability with increasing L-leu concentration. Therefore, this study highlighted that the relationship between flowability enhancement and morphological changes cannot be consistently predicted or generalised. 

Bulk characterisation data have previously shown that above a certain L-leu concentration, flowability improvements reached an optimum and plateaued [22,47], which is supported by our results. However, our results showed that the threshold concentration differed between the excipients. Two flowability tests showed this progression: shear cell and stability tests. Shear cell testing showed the behaviour of these powders in a confined high-stress environment, reminiscent of what could be observed in a hopper feed system [47]. PVP K10, maltodextrin, and trehalose exhibited this plateau behaviour in both the cohesion and ffc data. Finally, the AIF data did not provide substantial insights into the differences between formulations. 

The stability test represented the behaviour of these powders in the case of an unconfined powder flow. The BFE obtained from the standard stability test was insufficient to differentiate the formulations. Unlike the shear cell, BFE did not differentiate any formulation once L-leu had sufficiently minimised the interparticular cohesion (≤20 mJ). Therefore, we conclude that SE provides a more representative measure of the flow behaviour of a formulated powder under unconfined stress. As this test measured the blade leaving the powder bed, it did not consider the energy required to compact the bed. The results were normalised by sample mass, eliminating bulk density differences and giving a better account of powder cohesion. 

A comparison of both confined and unconfined data shows how these surface-modified powders perform differently under different consolidation conditions. For example, shear cell data showed that coprocessed maltodextrin achieved maximum flowability improvement at a lower L-leu concentration than PVP K10 (≥5 wt% vs. ≥10 wt%). This trend was reversed in the SE data, where PVP K10 performed better than maltodextrin, achieving an improvement plateau at ≥5 wt% compared to ≥10 wt%.

In addition, the FRI data showed how altering L-leu level coprocessing altered the flowrate sensitivity of the powder. This is attributed to the differences in packing behaviour and efficiency of each formulation. At a lower speed, greater sensitivity is shown; therefore, we propose that the slower application of force allowed more time and opportunity for the materials to consolidate and form a bulk resistance towards the blade, increasing the total energy measured. The FRI data showed that PVP K10 was not greatly affected by alterations in blade speed, whereas maltodextrin and trehalose exhibited substantial flowrate sensitivity as a function of blade speed. 

To further explore this observation, a modified stability test was used, where the blade speed was reduced from 100 mm/s to 60 mm/s. It showed better differentiation for each formulation at a lower speed. For example, at a faster speed, trehalose displayed the lowest BFE among all the excipients. However, in the lower speed regime, trehalose formulations plateaued at a BFE higher than most other excipients. In addition, the stability test was unable to differentiate maltodextrin formulations in the 10–30 wt% L-leu range, whereas this modified test could differentiate the powders. We believe that this provides an improved approach to enhance the characterisation and differentiation of these powders based on their flow-rate sensitivities. 

The permeability test provided further insights into the compactability and air entrapment efficiency of leucine coprocessed powders. The powders were evaluated under a range of normal-stress regimes. The greatest difference between L-leu modifications occurred in the higher-stress regime (15 kPa). In general, a higher L-leu concentration resulted in an increase in the pressure drop across the powder bed. We attributed this to the increased air resistance and entrapment to their surface state and morphologies, which allowed them to pack more efficiently when subjected to an external force. All of these bulk tests showed that there were multiple exceptions and variations in the generalised preconception that L-leu improves bulk powder flowability. 

The excipients maltodextrin, PVP K10, and trehalose were homogeneous in nature and exhibited substantial surface modification from L-leu coformulation. Gum arabic coprocessed with L-leu was the only excipient investigated with a heterogeneous nature, containing a mixture of polysaccharides and glycoproteins [51]. It displayed morphological progression, such as maltodextrin, blood cell-like structures when purely spray dried, and rugose morphologies with substantial (≥20 wt%) L-leu. There was also a gradual increase in flowability at relatively higher L-leu concentrations. It is understood that other amino acids have surface activity similar to that of L-leu; however, they do not display the same degree of flowability improvements [17]. Since gum arabic contains a substantial proportion of amino acids within its composition, it was speculated that this may lead to a degree of interruption in the formation of the historically reported L-leu shell. This example of gum arabic and L-leu formulations highlights the potential challenges and uncertainties in outcomes when coprocessed with heterogeneous formulations, with one or more APIs and other components, which may be expected in pharmaceutical and nutraceutical products. 

Larger macromolecule excipients, such as PVP K90 and HPMCs, displayed contrasting results when coprocessed. PVP K90 displayed severe incompatibility at low L-leu concentrations (2.5–10 wt%) with the formation of fibrous byproducts, which necessitates future investigation. However, observing the morphologies of the free powders showed that they exhibited the same morphologies as the PVP K10 powders. Furthermore, at ≥15 wt% L-leu, PVP K90 formulations had bulk characteristics similar to those of PVP K10. This indicated that the large difference in molecular size did not greatly impact the ability of L-leu to modify the powders formed from these excipients. 

The HPMC E5LV and K100M formulations displayed contrasting characteristics compared to the other excipients. Morphologically, most bulk flowability tests showed that L-leu had little to no effect on the particles formed. However, XRD analysis showed the presence of the characteristic reported 6° and 19° 2θ peak, associated with a crystalline L-leu surface shell. These peaks in the HPMC XRDs displayed a degree of broadening compared with the other formulations. This may be consistent with the differences in the L-leu formation and the alternative form on the surface. HPMC is a known viscosity-modifying material [44]. Feng et al. reported that viscosity may play a role in hindering the ability of L-leu to form a crystalline shell [16]. The HPMCs used were also the largest molecules examined here and therefore would have a higher Péclet number relative to L-leu. HPMC was also reported to have surface activity, where it was able to improve the flowability of various ASDs in common with L-leu [52]. Therefore, a combination of such factors may result in HPMC having a physico-chemical interaction with L-leu, preventing it from adequately forming a well-ordered crystalline structure. Data from this study showed the contextual interaction between L-leu’s mechanism of action and the different physico-chemical characteristics of the coformulated excipients. Furthermore, it highlighted how the different FT4 powder rheometer tests characterised the different physical properties of each formulation. 

This study demonstrated the impact of L-leu surface modification on the bulk flowability performance of excipients in spray-dried formulations, but it also raised questions regarding the underlying mechanisms of L-leu surface crystallisation. In future work, to gain a deeper understanding of these mechanisms, we propose that various spectroscopic techniques, such as Fourier-transform infrared spectroscopy, X-ray photoelectron spectroscopy, and time-of-flight secondary ion mass spectrometry, can be used to investigate the surface characteristics of the modified materials. Additional techniques, such as inverse gas chromatography, can also provide insight into the surface characteristics of these surface modified powders. Future studies may focus on gum arabic and HPMC formulations, as their interaction with L-leu may offer insights into their mechanism of action through their disruption. Future work should also incorporate model materials with additional studies to examine aspects such as chemical stability and dissolution to further explore the impact of L-leu surface modification on the behaviour of spray-dried formulations.

## 5. Conclusions

Overall, this study showed the effects of L-leu coprocessing via spray drying on the bulk powder characteristics of different excipient formulations. Effects, such as improved flow and altered compaction, are likely to lead to better control of postprocessability. The addition of L-leu substantially modified the performance of excipients, such as PVP K10, maltodextrin, and trehalose, which may help in the design of potential ASDs. However, we observed novel behaviours of other common excipients, such as PVP K90, HPMCs, and gum arabic, which exhibited differing behaviours. This highlights that L-leu behaviour in particle formation differs with the differing physico-chemical properties of the coformulated excipients and will not be directly predictable for all materials. This work highlights the need for a better understanding of its mechanistic behaviour in multicomponent systems, especially with increasing complexity of the composition. This also demonstrates the variability of results from different standard bulk characteristics of such formulated powders and the importance of suitable test design and control in identifying the nature of bulk character changes due to surface modification with a coexcipient, such as l-leu. 

## Figures and Tables

**Figure 1 pharmaceutics-15-01447-f001:**
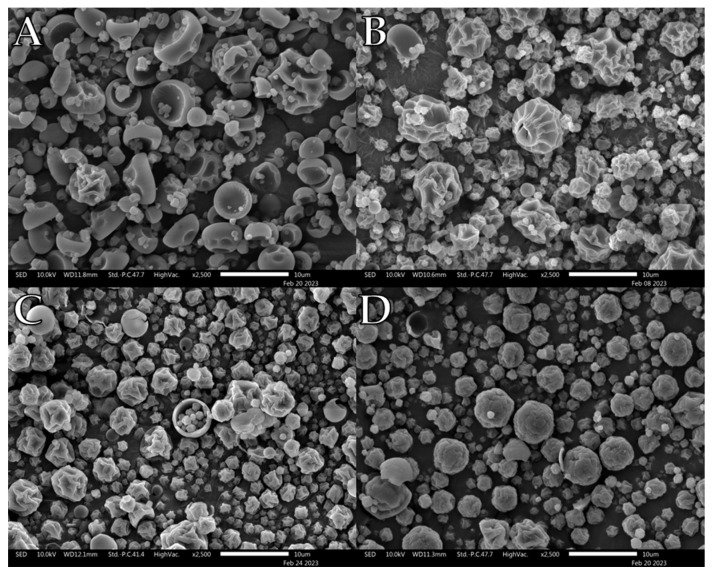
Representative SEM images of spray-dried maltodextrin/L-leu formulations: (**A**) MD/L-leu (0%), (**B**) MD/L-leu (5%), (**C**) MD/L-leu (10%), (**D**) MD/L-leu (20%).

**Figure 2 pharmaceutics-15-01447-f002:**
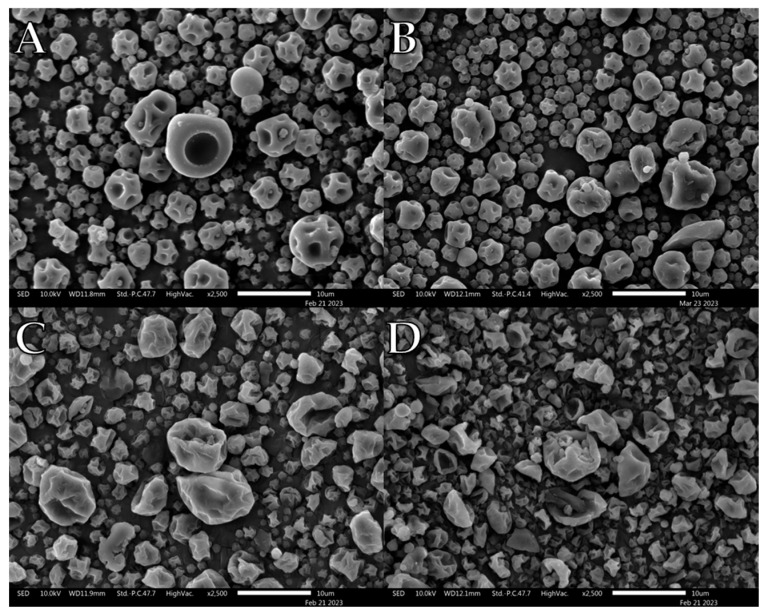
Representative SEM images of spray dried PVP K10/L-leu formulations: (**A**) PVP K10/L-leu (0%), (**B**) PVP K10/L-leu (5%), (**C**) PVP K10/L-leu (10%), (**D**) PVP K10/L-leu (20%).

**Figure 3 pharmaceutics-15-01447-f003:**
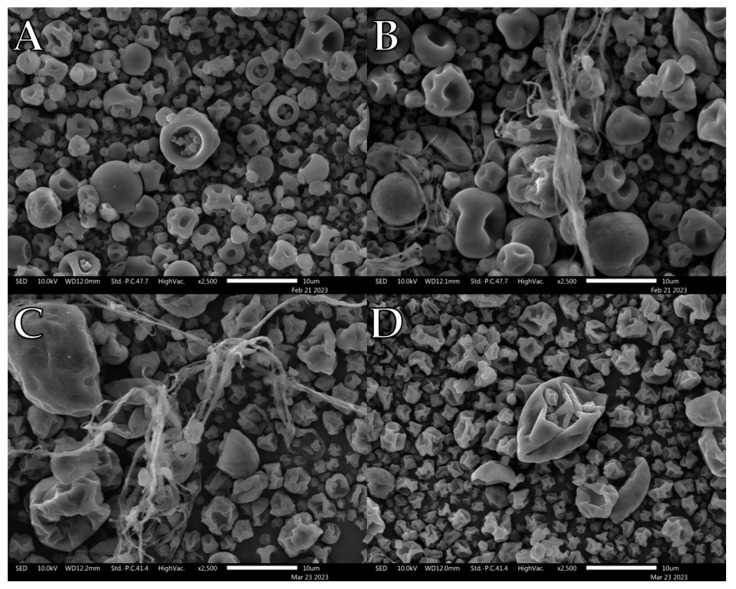
Representative SEM images of spray-dried PVP K90/L-leu formulations: (**A**) PVP K90/L-leu (0%), (**B**) PVP K90/L-leu (5%), (**C**) PVP K90/L-leu (10%), (**D**) PVP K90/L-leu (20%).

**Figure 4 pharmaceutics-15-01447-f004:**
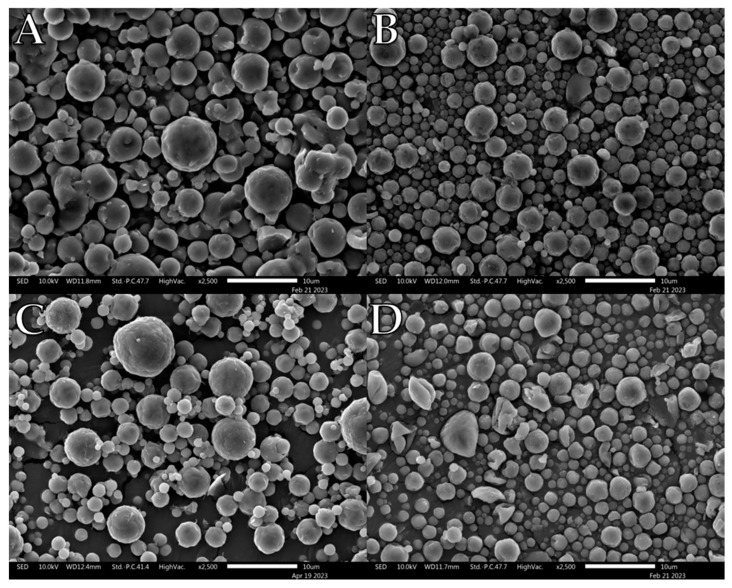
Representative SEM images of spray-dried trehalose/L-leu formulations: (**A**) Trh/L-leu (0%), (**B**) Trh/L-leu (5%), (**C**) Trh /L-leu (10%), (**D**) Trh /L-leu (20%).

**Figure 5 pharmaceutics-15-01447-f005:**
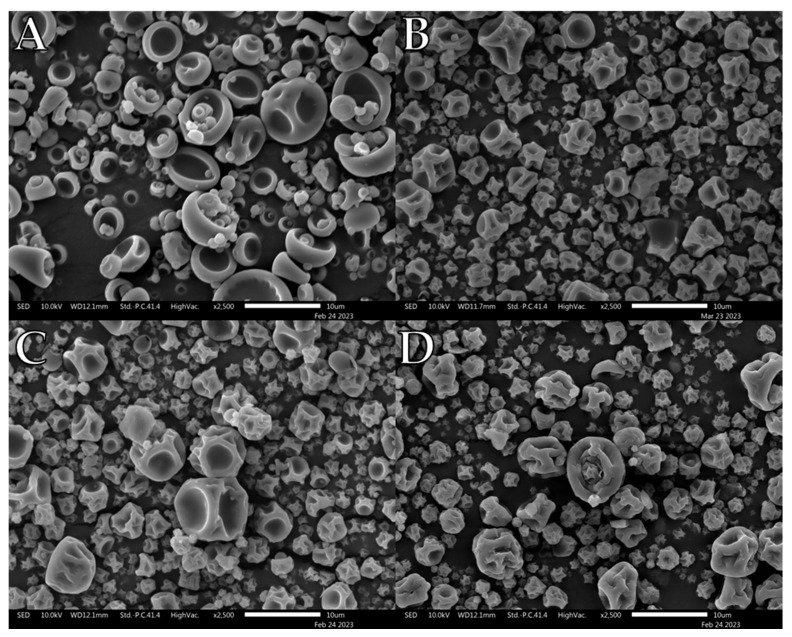
Representative SEM images of spray-dried gum arabic/L-leu formulations: (**A**) GA/L-leu (0%), (**B**) GA/L-leu (5%), (**C**) GA/L-leu (10%), (**D**) GA/L-leu (20%).

**Figure 6 pharmaceutics-15-01447-f006:**
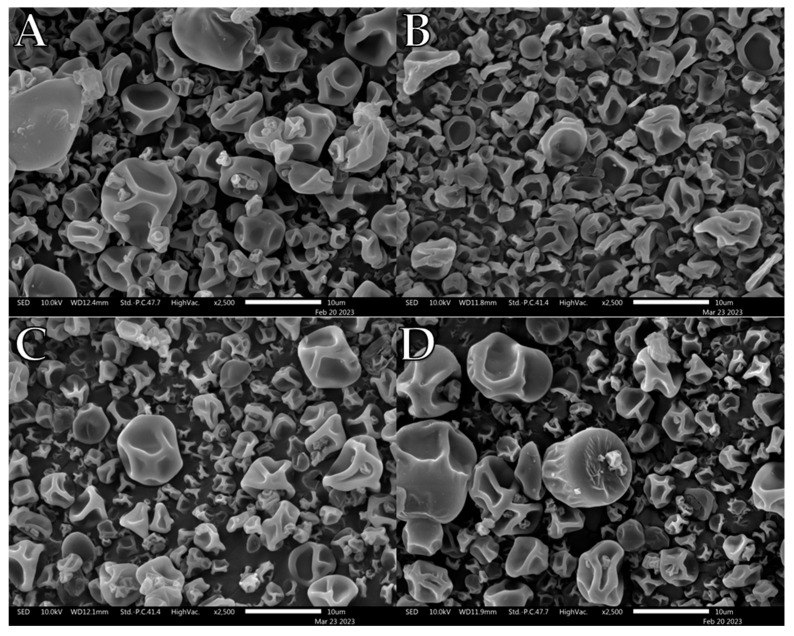
Representative SEM images of spray dried HPMC E5LV/L-leu formulations: (**A**) HPMC E5LV/L-leu (0%), (**B**) HPMC E5LV/L-leu (5%), (**C**) HPMC E5LV/L-leu (10%), (**D**) HPMC E5LV /L-leu (20%).

**Figure 7 pharmaceutics-15-01447-f007:**
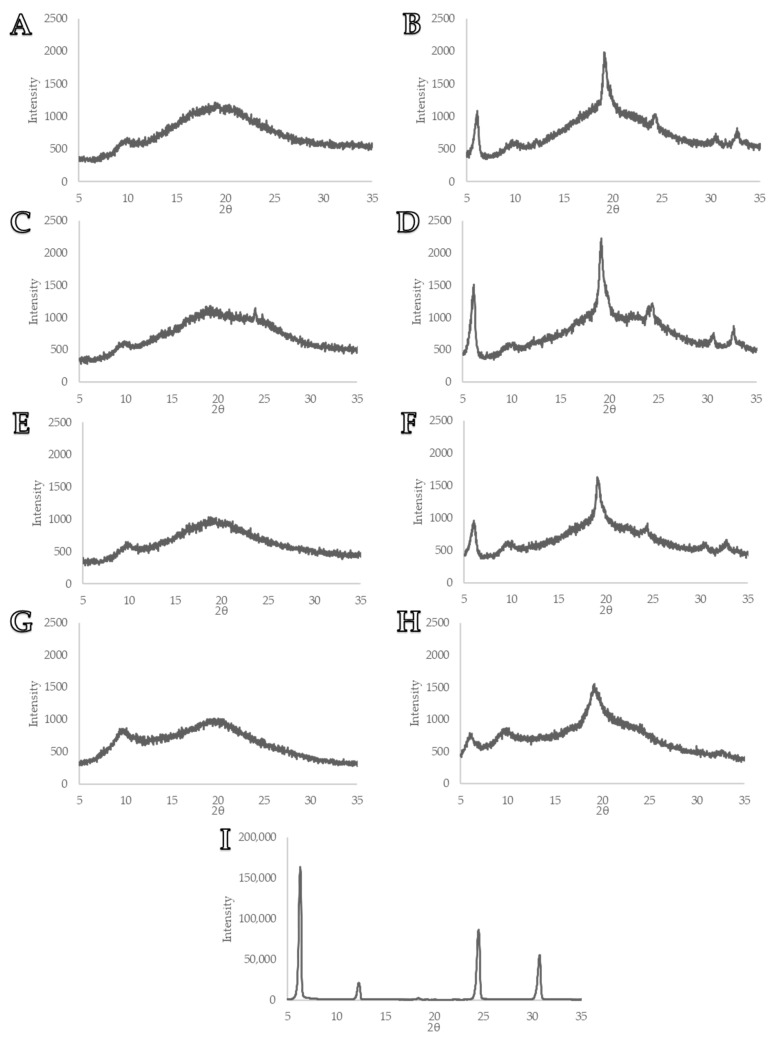
P-XRD diffractograms of various spray dried formulations: (**A**) MD/L-leu (0%), (**B**) MD/L-leu (20%) (**C**) Trh/L-leu (0%), (**D**) Trh/L-leu (20%), (**E**) GA/L-leu (0%), (**F**) GA/L-leu (20%), (**G**) HPMC E5LV/L-leu (0%), (**H**) HPMC E5LV/L-leu (20%), (**I**) Pure unspray-dried L-leu.

**Figure 8 pharmaceutics-15-01447-f008:**
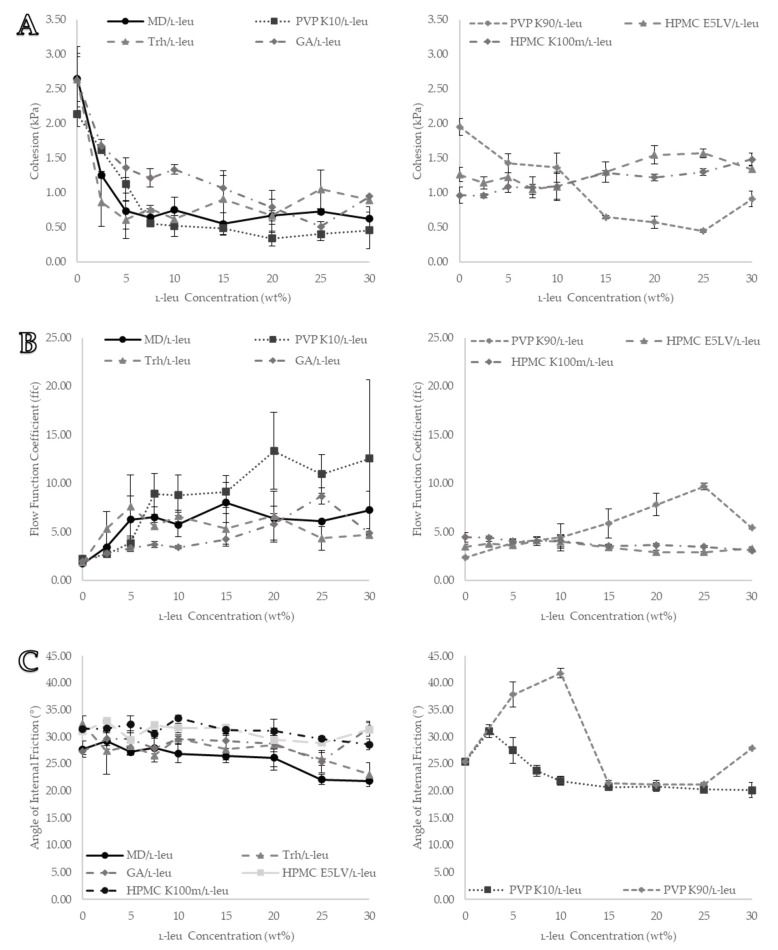
Visual representation of formulation shear cell data from the Freeman FT4 powder rheometer: (**A**) cohesion data, (**B**) flow function coefficient (ffc) data, and (**C**) angle of internal friction (AIF) data. All data points are displayed as mean ± SD (n = 3).

**Figure 9 pharmaceutics-15-01447-f009:**
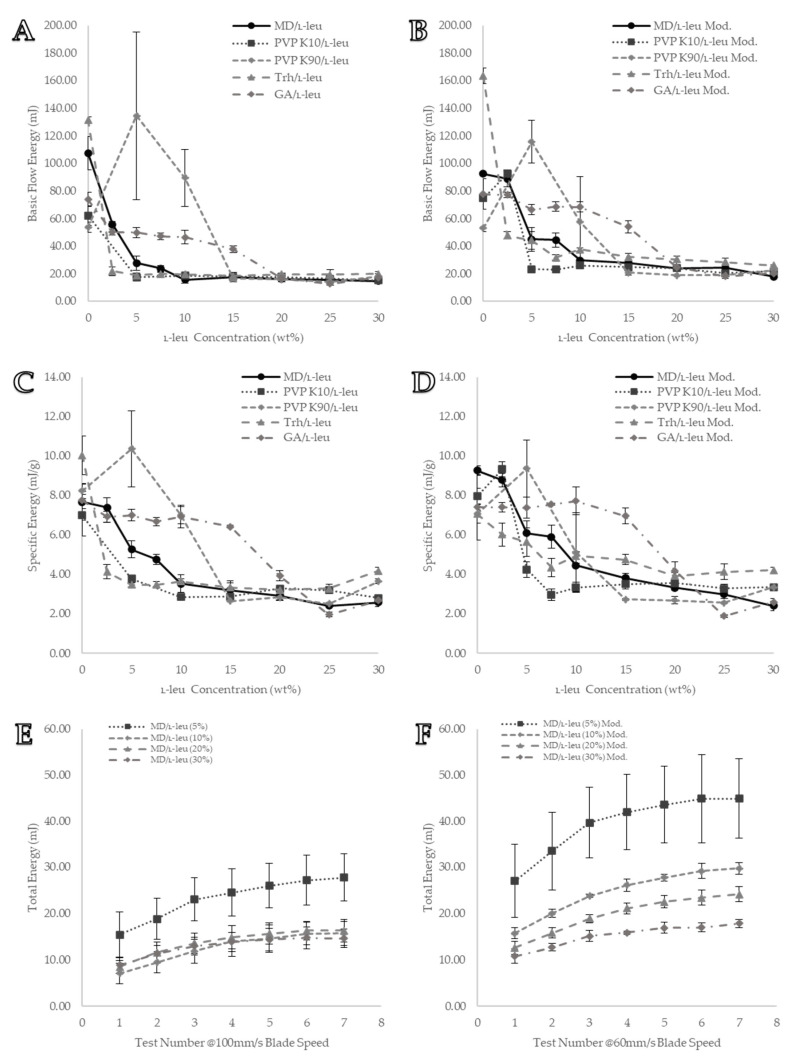
Visual representation of formulation stability data from the Freeman FT4 powder rheometer: (**A**) BFE value derived from the standard protocol; (**B**) BFE value derived from the modified protocol; (**C**) SE value derived from the standard protocol; (**D**) SE value derived from the modified protocol; (**E**) MD/L-leu stability data from the standard protocol; (**F**) MD/L-leu stability data from the modified protocol. All data points are displayed as mean ± SD (n = 3).

**Figure 10 pharmaceutics-15-01447-f010:**
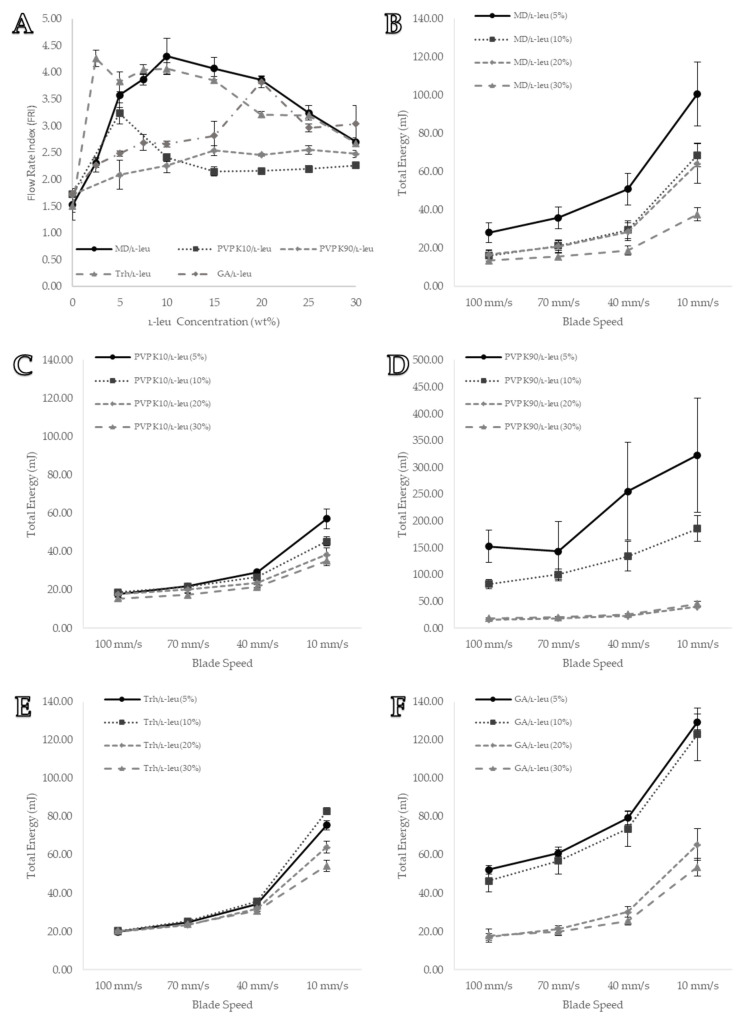
Visual representation of formulation variable flow rate data from the Freeman FT4 powder rheometer: (**A**) FRI values of each formulation tested; (**B**) MD/L-leu powder FRI data; (**C**) PVP K10/L-leu powder FRI data; (**D**) PVP K90/L-leu powder FRI data; (**E**) Trh/L-leu powder FRI data; and (**F**) GA/L-leu powder FRI data. All data points are displayed as mean ± SD (n = 3).

**Figure 11 pharmaceutics-15-01447-f011:**
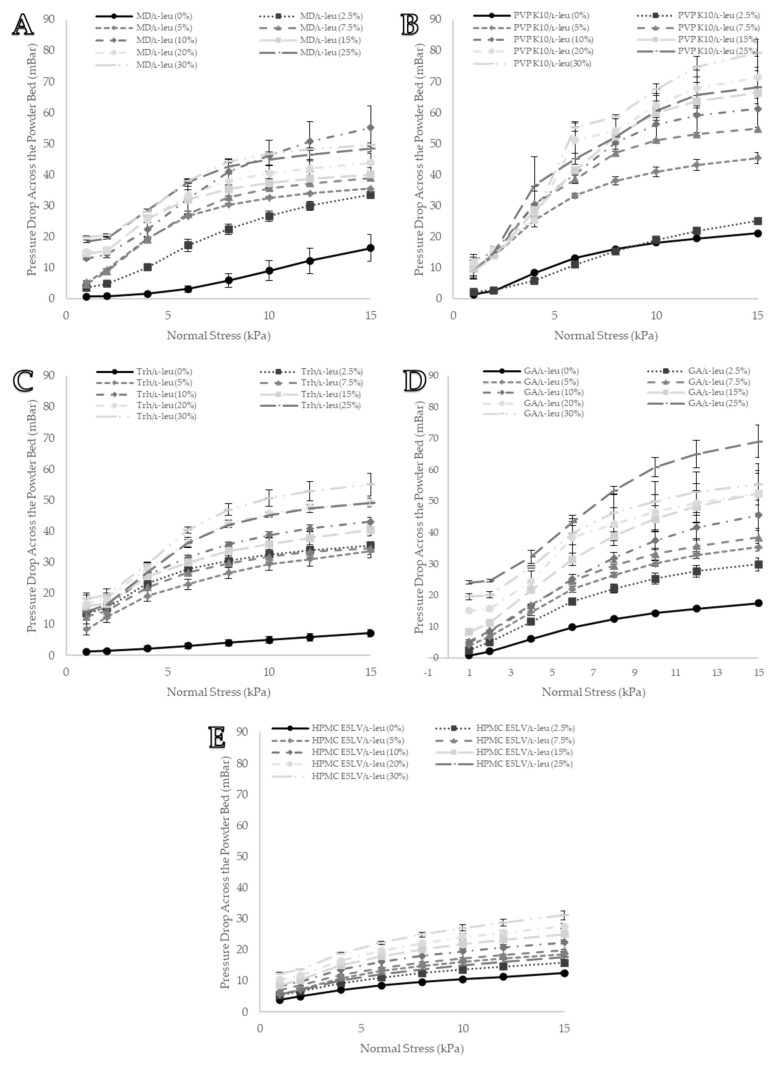
Visual representation of the formulation permeability data from the Freeman FT4 powder rheometer: (**A**) MD/L-leu powder permeability data, (**B**) PVP K10/L-leu powder permeability data, (**C**) Trh/L-leu powder permeability data, (**D**) GA/L-leu powder permeability data, and (**E**) HPMC E5LV/L-leu powder permeability data. All data points are displayed as mean ± SD (n = 3).

**Table 1 pharmaceutics-15-01447-t001:** List of formulation compositions for spray drying.

Excipient	Abbreviation	Formulation (Excipient/L-Leu Mixture)	L-leucine Content Percentage (L-Leu wt% Dry Basis)
Maltodextrin	MD	MD/L-leu	0, 2.5, 5, 7.5, 10, 15, 20, 30%
Polyvinylpyrrolidone K10	PVP K10	PVP K10/L-leu	0, 2.5, 5, 7.5, 10, 15, 20, 30%
Polyvinylpyrrolidone K90	PVP K90	PVP K90/L-leu	0, 2.5, 5, 7.5, 10, 15, 20, 30%
Trehalose	Trh	Trh/L-leu	0, 2.5, 5, 7.5, 10, 15, 20, 30%
Gum Arabic	GA	GA/L-leu	0, 2.5, 5, 7.5, 10, 15, 20, 30%
Hydroxypropyl methylcellulose E5LV	HPMC E5LV	HPMC E5LV/L-leu	0, 2.5, 5, 7.5, 10, 15, 20, 30%
Hydroxypropyl methylcellulose K100M	HPMC K100M	HPMC K100M/L-leu	0, 2.5, 5, 7.5, 10, 15, 20, 30%

**Table 2 pharmaceutics-15-01447-t002:** Particle size data of the spray-dried formulations. Data represent the mean ± SD (*n* = 3).

Formulation Code	D10 (µm)	D50 (µm)	D90 (µm)	Span	Formulation Code	D10 (µm)	D50 (µm)	D90 (µm)	Span
MD/L-leu (0%)	3.17 ± 0.01	4.69 ± 0.03	8.79 ± 0.42	1.2 ± 0.15	GA/L-leu (0%)	3.78 ± 0.02	6.55 ± 0.04	11.61 ± 0.21	1.2 ± 0.09
MD/L-leu (2.5%)	3.55 ± 0.01	5.94 ± 0.01	11.07 ± 0.25	1.27 ± 0.09	GA/L-leu (2.5%)	3.74 ± 0.01	6.52 ± 0.02	11.54 ± 0.09	1.2 ± 0.04
MD/L-leu (5%)	3.14 ± 0.02	4.72 ± 0.01	8.72 ± 0.11	1.18 ± 0.04	GA/L-leu (5%)	3.67 ± 0.03	6.17 ± 0.07	10.76 ± 0.15	1.15 ± 0.08
MD/L-leu (7.5%)	3.49 ± 0.01	5.62 ± 0.07	10.2 ± 0.11	1.2 ± 0.06	GA/L-leu (7.5%)	3.78 ± 0.02	6.57 ± 0.06	11.27 ± 0.14	1.14 ± 0.08
MD/L-leu (10%)	3.52 ± 0.01	5.83 ± 0.05	10.46 ± 0.21	1.19 ± 0.09	GA/L-leu (10%)	3.48 ± 0.03	5.38 ± 0.06	9.55 ± 0.08	1.13 ± 0.05
MD/L-leu (15%)	2.92 ± 0	4.49 ± 0.01	7.96 ± 0.13	1.12 ± 0.05	GA/L-leu (15%)	3.49 ± 0.01	5.45 ± 0.01	9.66 ± 0.06	1.13 ± 0.03
MD/L-leu (20%)	2.95 ± 0.01	4.54 ± 0.02	8.47 ± 0.26	1.21 ± 0.09	GA/L-leu (20%)	3.58 ± 0.01	6.28 ± 0.03	11.23 ± 0.13	1.22 ± 0.05
MD/L-leu (25%)	3.08 ± 0.04	4.71 ± 0.01	9.23 ± 0.04	1.31 ± 0.03	GA/L-leu (25%)	3.42 ± 0.01	5.4 ± 0.02	10.23 ± 0.08	1.26 ± 0.04
MD/L-leu (30%)	3.42 ± 0.02	5.78 ± 0.11	10.87 ± 0.22	1.29 ± 0.12	GA/L-leu (30%)	3.66 ± 0.02	6.63 ± 0.02	16.38 ± 0.79	1.92 ± 0.27
PVP K10/L-leu (0%)	3.69 ± 0.02	6.12 ± 0.06	10.38 ± 0.1	1.09 ± 0.06	HPMC E5LV/L-leu (0%)	4.05 ± 0.03	8.1 ± 0.14	17.77 ± 0.69	1.69 ± 0.29
PVP K10/L-leu (2.5%)	3.49 ± 0.02	5.48 ± 0.04	9.84 ± 0.05	1.16 ± 0.03	HPMC E5LV/L-leu (2.5%)	4 ± 0.02	7.95 ± 0.06	17.01 ± 0.54	1.63 ± 0.21
PVP K10/L-leu (5%)	3.19 ± 0	4.75 ± 0.02	8.88 ± 0.07	1.2 ± 0.03	HPMC E5LV/L-leu (5%)	4.18 ± 0.01	9.13 ± 0.1	19.74 ± 0.71	1.7 ± 0.27
PVP K10/L-leu (7.5%)	3.67 ± 0.03	6.13 ± 0.02	10.4 ± 0.06	1.1 ± 0.04	HPMC E5LV/L-leu (7.5%)	4.16 ± 0.06	9.09 ± 0.13	22.11 ± 0.94	1.98 ± 0.38
PVP K10/L-leu (10%)	3.41 ± 0.02	5.02 ± 0.05	8.98 ± 0.08	1.11 ± 0.05	HPMC E5LV/L-leu (10%)	3.93 ± 0.02	7.77 ± 0.07	17.64 ± 0.59	1.76 ± 0.23
PVP K10/L-leu (15%)	3.37 ± 0.01	5.01 ± 0.04	9.01 ± 0.06	1.12 ± 0.04	HPMC E5LV/L-leu (15%)	4.08 ± 0.01	8.33 ± 0.07	18.82 ± 0.76	1.77 ± 0.28
PVP K10/L-leu (20%)	3.42 ± 0.01	5.02 ± 0.04	9.04 ± 0.14	1.12 ± 0.06	HPMC E5LV/L-leu (20%)	3.95 ± 0.02	7.91 ± 0.1	17.43 ± 0.42	1.7 ± 0.18
PVP K10/L-leu (25%)	3.29 ± 0.01	4.75 ± 0.01	8.47 ± 0.12	1.09 ± 0.05	HPMC E5LV/L-leu (25%)	3.97 ± 0.01	7.84 ± 0.01	17.93 ± 0.29	1.78 ± 0.1
PVP K10/L-leu (30%)	3.39 ± 0.02	4.88 ± 0.02	8.75 ± 0.08	1.1 ± 0.04	HPMC E5LV/L-leu (30%)	3.89 ± 0.03	7.57 ± 0.12	17.66 ± 1.35	1.82 ± 0.5
Trh/L-leu (0%)	3.77 ± 0.09	6.21 ± 0.15	12.64 ± 0.54	1.43 ± 0.26	HPMC K100M/L-leu (0%)	4.29 ± 0.01	11.43 ± 0.15	26.4 ± 0.49	1.93 ± 0.22
Trh/L-leu (2.5%)	3.57 ± 0.04	6.06 ± 0.07	11.93 ± 0.48	1.38 ± 0.19	HPMC K100M/L-leu (2.5%)	4.39 ± 0.02	11.01 ± 0.17	24.97 ± 0.48	1.87 ± 0.22
Trh/L-leu (5%)	3.15 ± 0.03	4.72 ± 0.04	9.71 ± 0.53	1.39 ± 0.2	HPMC K100M/L-leu (5%)	3.94 ± 0.01	8.71 ± 0.11	20.79 ± 0.33	1.93 ± 0.15
Trh/L-leu (7.5%)	3.13 ± 0.06	4.68 ± 0.04	8.99 ± 0.14	1.25 ± 0.08	HPMC K100M/L-leu (7.5%)	4.29 ± 0.02	10.75 ± 0.3	25.28 ± 1.34	1.95 ± 0.55
Trh/L-leu (10%)	3.14 ± 0.02	4.72 ± 0.01	9.82 ± 0.22	1.42 ± 0.08	HPMC K100M/L-leu (10%)	3.76 ± 0.02	8.38 ± 0.19	20.99 ± 0.95	2.06 ± 0.39
Trh/L-leu (15%)	3.07 ± 0.11	4.88 ± 0.36	13.33 ± 4.13	2.1 ± 1.53	HPMC K100M/L-leu (15%)	4.1 ± 0.02	9.82 ± 0.22	24.12 ± 0.81	2.04 ± 0.35
Trh/L-leu (20%)	3.36 ± 0.03	5.28 ± 0.03	11.11 ± 0.18	1.47 ± 0.08	HPMC K100M/L-leu (20%)	4.07 ± 0.03	8.99 ± 0.11	20.62 ± 1.3	1.84 ± 0.48
Trh/L-leu (25%)	3.64 ± 0.03	6.37 ± 0.06	12.49 ± 0.24	1.39 ± 0.11	HPMC K100M/L-leu (25%)	4.14 ± 0.02	9.49 ± 0.09	21.62 ± 0.47	1.84 ± 0.19
Trh/L-leu (30%)	3.76 ± 0.03	6.37 ± 0.05	11.07 ± 0.06	1.15 ± 0.04	HPMC K100M/L-leu (30%)	4.09 ± 0.02	9.08 ± 0.13	20.11 ± 0.22	1.76 ± 0.12
PVP K90/L-leu (0%)	3.96 ± 0.03	7.99 ± 0.07	16.3 ± 0.12	1.54 ± 0.07					
PVP K90/L-leu (5%)	4.32 ± 0.04	8.93 ± 0.04	18.13 ± 0.31	1.55 ± 0.13					
PVP K90/L-leu (10%)	4.06 ± 0.66	8.96 ± 0.62	18.41 ± 0.91	1.6 ± 0.73					
PVP K90/L-leu (15%)	3.98 ± 0.01	8.19 ± 0.02	16.18 ± 0.15	1.49 ± 0.06					
PVP K90/L-leu (20%)	3.98 ± 0.02	8 ± 0.04	15.72 ± 0.23	1.47 ± 0.09					
PVP K90/L-leu (25%)	4.16 ± 0.02	8.89 ± 0.11	18.12 ± 0.53	1.57 ± 0.22					
PVP K90/L-leu (30%)	4.09 ± 0.02	8.27 ± 0.05	16.41 ± 0.53	1.49 ± 0.2					

**Table 3 pharmaceutics-15-01447-t003:** Guidelines correlating ffc values with powder flow characteristics.

Flow Function Coefficient	
Ratio	Flowability
<2	Very Cohesive
2–4	Cohesive
4–10	Easy-flowing
>10	Free-flowing

## Data Availability

Not applicable.

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
