# Peer review of "Bulk Flow Optimisation of Amorphous Solid Dispersion Excipient Powders through Surface Modification"

_pharmaceutics, 2023, doi:10.3390/pharmaceutics15051447_

Round 1

Reviewer 1 Report

The manuscript is well-written, experiments are well designed, results and discussions are sound.  The work provides a good in-depth assessment of the impact of leucine on bulk material properties of spray dried excipients commonly used in pharmaceuticals. However, there are several minor grammatical errors. For instance, authors use "was" when describing or defining properties of excipients. Please find below other comments which authors should address.

1. Authors allude in the abstract that the inherent nature of most ASDs leads to surface cohesion/adhesion which hinders their bulk flow. However this dependents on the ASD manufacturing technique. Example ASDs manufactured by HME followed by milling of the extrudates into powder do not suffer this drawback. Please revise accordingly.

2. Line109-120: Please rewrite for clarity and check grammar.

3. Line 141: Define the abbreviation TDS

3. Level of residual solvent in spray dried formulation can affect its bulk material properties. What was the residual solvent of the formulations after manufacturing? Was secondary drying performed?

Line 362-363: How can such a conclusion regarding internal structure be drawn based on the P-XRD diffractogram? Further clarification is needed.

There are several minor grammatical errors

Author Response

Thank you for your time and effort in reviewing this manuscript. We very much appreciate the input from your experience and insights in providing valuable comments and requested modifications, which have helped us improve the quality of our manuscript. We are pleased to inform you that we believe we have addressed your concerns and have revised the manuscript accordingly. Please find attached the revised manuscript, along with a detailed response to your comments. We have addressed your review point-by-point as follows:

  1. Authors allude in the abstract that the inherent nature of most ASDs leads to surface cohesion/adhesion which hinders their bulk flow. However this dependents on the ASD manufacturing technique. Example ASDs manufactured by HME followed by milling of the extrudates into powder do not suffer this drawback. Please revise accordingly.
  • This is a good point, the term ‘most’ has been replaced with ‘spray-dried’ powders to enhance the focus of this study.
  1. Line109-120: Please rewrite for clarity and check grammar.
  • This section has been rewritten to better explain the differences between the HPMC grades. Grammar was checked and corrected extensively throughout the entire manuscript.
  1. Line 141: Define the abbreviation TDS
  • Thanks, very good point and this oversight was corrected, and the proper term total dissolved solids was used in lieu of its abbreviation.
  1. Level of residual solvent in spray dried formulation can affect its bulk material properties. What was the residual solvent of the formulations after manufacturing? Was secondary drying performed?
  • This was an excellent point made by the reviewer. We acknowledge that the relative differences in moisture content play a substantial role in altering the bulk properties of the powders. No subsequent secondary drying was performed on any of the spray-dried samples. However, they were stored in a sealed, controlled low-humidity environment to minimise moisture uptake. Future works will explore the impact that relative moisture content has on powder flowability and ÊŸ-leucine effect on powder capillary interactions.
  1. How can such a conclusion regarding internal structure be drawn based on the P-XRD diffractogram? Further clarification is needed.
  • Conclusions regarding the internal structure of the ÊŸ-leucine surface coating were not made based on the P-XRD data from this study. Instead, it was a reiteration of investigations done in previous publications (doi:10.1016/j.ejpb.2012.10.015). This section has been reworked to alleviate any potential confusion.

We believe that our revised manuscript is now more comprehensive and will make a valuable contribution to this field. We would like to thank you for their guidance and recommendations, which have undoubtedly improved the quality of our manuscript.

Thank you again for your time and effort.

Reviewer 2 Report

Sadly, from many perspectives, this manuscript is difficult to read and understand. Numerous mistakes have been made in the writing, conceptualization and delivery of the potential publication.

The only mitigating “saviour” in the manuscript is an acceptable amount of new scientific data relating to the important, contemporary experimental subject matter discussed by the authors.

1.       The manuscript is, unfortunately, totally unacceptable in terms of grammar/syntax, and amongst many other factors, unnecessary capitalization of terms, incorrect phraseology, not to mention the incorrect use of abbreviations and tense used throughout this potential publication.

2.       Moreover, besides being “badly” written, the scientific information provided does not follow a logical, sequential pathway.

3.       Why do the authors not categorically state, early in the manuscript, as to why a hydrophobic amino acid was used? Indeed why L-Leu and not another hydrophobic amino acid (there is after all a substantive amount of data in the scientific literature relating to L-Leu for the purpose for which this amino acid has been employed in their studies)?

4.       I would recommend that the authors undertake a complete revision of the manuscript taking into account the above comments. In addition the following is advised.

(i)                  Refer to the following two manuscripts: Pharmaceutics 2023, 15, 435. https://doi.org/10.3390/pharmaceutics15020435 and Pharmaceutical Research (2022) 39:3047–3061.

(ii)                In the discussion section, the authors need to think about suggesting the use of other techniques/methodologies which may help/enhance in the interpretation of their data. The foregoing include, but are not limited to DSC, TGA, DVS, XPS, ToF-SIMS and particle size determination using e.g.,  laser diffraction .

See comments to the authors

Author Response

Thank you for your time and effort in reviewing this manuscript. We very much appreciate the input from your experience and insights in providing valuable comments and requested modifications, which have helped us improve the quality of our manuscript. We are pleased to inform you that we believe we have addressed your concerns and have revised the manuscript accordingly. Please find attached the revised manuscript, along with a detailed response to your comments. We have addressed your review point-by-point as follows:

  1. The manuscript is, unfortunately, totally unacceptable in terms of grammar/syntax, and amongst many other factors, unnecessary capitalization of terms, incorrect phraseology, not to mention the incorrect use of abbreviations and tense used throughout this potential publication.
  • The authors very much appreciate the extent and depth of review and fully acknowledge that a significant reworking of the manuscript was required to address points. We conducted a comprehensive re-writing of the manuscript accordingly to correct all grammar, capitalisation, and phraseology. We have corrected the phraseology and abbreviation used in this paper to make it more appropriate. We believe it is now suitable for publication as a result.
  1. Moreover, besides being “badly” written, the scientific information provided does not follow a logical, sequential pathway.

  • The authors thank the reviewer for this important comment and we worked through clarifying the logic of the material presented, a major revision of the introduction has clarified the organisation of the manuscript. The data are presented to highlight the rational progression that occurred during this study. Initial datasets, such as qualitative observations of powder physical behaviour and morphological characteristics supplemented by P-XRD, confirmed what had been shown in previous publications; ÊŸ-leucine alters the surface of spray-dried powders by forming a crystalline surface layer. This was followed by shear cell testing, which is the gold standard approach for characterising bulk flowability in particle engineering space. Shear cell data provided the necessary baseline data for comparison with more novel characterisation techniques available in Freeman FT4, such as stability testing, variable flowrate testing, and permeability testing.

  1. Why do the authors not categorically state, early in the manuscript, as to why a hydrophobic amino acid was used? Indeed why L-Leu and not another hydrophobic amino acid (there is after all a substantive amount of data in the scientific literature relating to L-Leu for the purpose for which this amino acid has been employed in their studies)?

  • The authors acknowledge this deficiency; the introduction has been amended to include a section describing the historical rationale for choosing ÊŸ-leucine over other amino acids with hydrophobic side chains.

  1. I would recommend that the authors undertake a complete revision of the manuscript taking into account the above comments. In addition the following is advised.

(i) Refer to the following two manuscripts: Pharmaceutics 2023, 15, 435. https://doi.org/10.3390/pharmaceutics15020435 and Pharmaceutical Research (2022) 39:3047–3061.

  • The authors would like to thank the reviewer for these manuscript recommendations and have provided excellent background information on the mechanistic properties of ÊŸ-leucine. Subsequently, the insights provided have been included in the manuscript and appropriately referenced.

(ii) In the discussion section, the authors need to think about suggesting the use of other techniques/methodologies which may help/enhance in the interpretation of their data. The foregoing include, but are not limited to DSC, TGA, DVS, XPS, ToF-SIMS and particle size determination using e.g., laser diffraction .

  • We appreciate the reviewer for recommending the use of additional spectroscopic techniques to improve the interpretation of the mechanistic reason behind ÊŸ-leucine activity. We were already conducting work in this area as a follow-up study, and note it is important that the current paper is directed purely at bulk powder characterisation. The Discussion section has been updated to incorporate this important clarification and the use of these techniques in future investigations.

We believe that our substantially revised manuscript is now more comprehensive and will make a valuable contribution to this field. We would like to thank you for their guidance and recommendations, which have undoubtedly improved the quality of our manuscript.

Thank you again for your time and effort.

Reviewer 3 Report

1. In the abstract, you state the abbreviations that you did not introduce. You use PVP, and there is a lot of Polyvinylpyrrolidone on the market.

2. Please check in the Instructions whether the references are given [1][2] or [1, 2] and correct this throughout the paper.

3. Row 53, do you think that the addition of magnesium stearate, as an excellent lubricant, and agent for improving sliding and flow, would affect the reduction of solubility and dissolution rate of sparingly soluble medicinal substances, which is certainly the case with substances in the formulation of solid dispersions. The goal of formulating solid dispersions is to increase solubility/dissolution rate, and magnesium stearate reduces this.

4. Line 68, consider using co-excipient or excipient instead of the word co-additive, which is more commonly used in the food industry

5. In line 101 there is no PVP, those abbreviations and numbers must be mentioned throughout the paper

6. For each excipient when it is introduced for the first time, the chemical name and the pharmacopoeial name must be specified, if the excipient has pharmacopoeial status, and only then the protected name that will be used in its entirety later in the work date.

7. The markings in Table 1 with xx are absolutely confusing, think about it

8. I am just now registering, I would like you to refute me, that the title has nothing to do with the work! You mention solid dispersions in the title. Solid dispersions are by definition a mixture of a medicinal substance in a suitable matrix carrier (excipient). By definition they are made with a healing substance, I don't see a healing substance here. I have the impression that you have here a formulation of co-processed excipients for the production of solid dispersions, in order to improve the characteristics of the matrix, but also of solid dispersions.

9. Chapter 2.4. is absolutely unclear, here you are already starting to describe the results. Separate methods and results by substance. Please reword this chapter and make it clearer. For equation one, for each of the parameters, state what it is. For example, lines 174 and 175 seem completely unclear. Is there any reference for this procedure? Find out, I wouldn't tell you that it turns out that I favor some authors in quoting.

10. I ask the same for chapter 2.6. references. I'm slowly getting lost with abbreviations, and so will the reader. What is FT4?

11. References for all procedures are missing and I will not list that again

12. Line 241 Spaces between numbers

13. Pictures 1-6 show absolutely no magnification, axis, or dimensions. Please correct the image.

14. Labels in Table 2 some first introduced vision. From one of my previous questions, it is not clear to me what D10, D50, D90, and Span are. If Span is a surfactant, it must be previously mentioned, although maybe it is and I did not register it, and on the other hand, if it is a surfactant, it must have a number.

15. I have to live here, perhaps it should have been at the end, what is the goal here, the co-processed excipient, do we think about how it will affect the rate of fermentation in the final formulation and the dissolution of the medicinal substance. Whether our goal is perhaps the use of lipid excipients, which can have a positive effect on the dissolution rate, must be clarified. But the title is not clear. This should be clarified in the introduction and reinforced throughout the work.

16. 3.2. chapter little discussed

17. I don't like the word formulation. This means the medicinal substance. Co-processed excipients.

18. Figure 7 units on both the x-axis and the y-axis. Pay attention to the discrete peaks that are observed, for example in the ACEG images around 10 degrees. The general shape is like this because of a lot of amorphous material, but if there is nothing crystalline, look at the more detailed diffractograms and potential, common, significant peaks.

19. Think again about the term wording

20. See for yourself the non-uniformity in Figure 10.

21. Check the results for some pictures, for example, 11f, there are illogical points, or explain them and support them with literature

22. Check references.

There are mistakes in the English language. Non-uniformity bothers me. The formulation and the co-processed excipient are mixed. The title is not good. Abbreviations are confusing.

Author Response

Thank you for your time and effort in reviewing this manuscript. We very much appreciate the input from your experience and insights in providing valuable comments and requested modifications, which have helped us improve the quality of our manuscript. We are pleased to inform you that we believe we have addressed your concerns and have revised the manuscript accordingly. Please find attached the revised manuscript, along with a detailed response to your comments. We have addressed your review point-by-point as follows:

  1. In the abstract, you state the abbreviations that you did not introduce. You use PVP, and there is a lot of Polyvinylpyrrolidone on the market.
  • This oversight has been rectified and the abbreviation has been replaced with the full chemical name and used grades in the abstract.
  1. Please check in the Instructions whether the references are given [1][2] or [1, 2] and correct this throughout the paper.
  • The authors appreciate the reviewers’ keen eye for details, and this issue has been resolved throughout the manuscript.
  1. Row 53, do you think that the addition of magnesium stearate, as an excellent lubricant, and agent for improving sliding and flow, would affect the reduction of solubility and dissolution rate of sparingly soluble medicinal substances, which is certainly the case with substances in the formulation of solid dispersions. The goal of formulating solid dispersions is to increase solubility/dissolution rate, and magnesium stearate reduces this.
  • The authors acknowledge this very important technical aspect, and as a result we have added the description of magnesium stearate surface coating potentially reducing dissolution rates in the introduction. We note the exploration of the degree to which magnesium stearate hinders dissolution when coated is outside the scope of this study. So, we believe this further clarification is sufficient.
  1. Line 68, consider using co-excipient or excipient instead of the word co-additive, which is more commonly used in the food industry.
  • Manuscript phraseology amended, replaced the term ‘co-additive’ with ‘co-excipient’.
  1. In line 101 there is no PVP, those abbreviations and numbers must be mentioned throughout the paper
  • Line amended to include PVP.
  1. For each excipient when it is introduced for the first time, the chemical name and the pharmacopoeial name must be specified, if the excipient has pharmacopoeial status, and only then the protected name that will be used in its entirety later in the work date.
  • This recommendation has been acknowledged and implemented in this manuscript.
  1. The markings in Table 1 with xx are absolutely confusing, think about it
  • Table 1 has been completely replaced with new terminology for the formulations to correct the confusion identified here.
  1. I am just now registering, I would like you to refute me, that the title has nothing to do with the work! You mention solid dispersions in the title. Solid dispersions are by definition a mixture of a medicinal substance in a suitable matrix carrier (excipient). By definition they are made with a healing substance, I don't see a healing substance here. I have the impression that you have here a formulation of co-processed excipients for the production of solid dispersions, in order to improve the characteristics of the matrix, but also of solid dispersions.
  • Introduction has been extensively amended to clarify the goal of this study was the synthesis and evaluation of delivery vehicle powders using known amorphous forming excipients as an approach to prototype solid dispersions as opposed to true solid dispersions themselves. The authors acknowledge that the introduction of bioactive ingredients can affect the performance of delivery vehicles. Therefore, it is important to generate baseline flowability data of pure carriers and ÊŸ-leucine for comparison with planned future studies which will incorporate active pharmaceutical ingredients.
  1. Chapter 2.4. is absolutely unclear, here you are already starting to describe the results. Separate methods and results by substance. Please reword this chapter and make it clearer. For equation one, for each of the parameters, state what it is. For example, lines 174 and 175 seem completely unclear. Is there any reference for this procedure? Find out, I wouldn't tell you that it turns out that I favor some authors in quoting.
  • Appreciate this important point, and manuscript has been amended with the lines describing the use of pressure titrations to determine the optimal dispersion pressure for particle sizing have been moved to the results section to alleviate this issue.
  1. I ask the same for chapter 2.6. references. I'm slowly getting lost with abbreviations, and so will the reader. What is FT4?
  • We appreciate picking this up, we have clarified in the manuscript that FT4 refers to the Freeman FT4 Rheometer; this abbreviation is the stated nomenclature used by the manufacturing company and widely employed across the literature.
  1. References for all procedures are missing and I will not list that again
  • References to operations manuals have been included in the methods section, wherever relevant.
  1. Line 241 Spaces between numbers
  • Spaces added between numbers.
  1. Pictures 1-6 show absolutely no magnification, axis, or dimensions. Please correct the image.
  • Thank you for the point, this was a formatting issue which is now resolved, and the relevant microscopy settings and magnification values are now visible.
  1. Labels in Table 2 some first introduced vision. From one of my previous questions, it is not clear to me what D10, D50, D90, and Span are. If Span is a surfactant, it must be mentioned previously, although maybe it is and I did not register it, and on the other hand, if it is a surfactant, it must have a number.
  • The authors thank the reviewer for this potential confusion, we can clarify that D10, D50, D90, and Span refer to a particle size statistic measure for particle size distributions. D10, D50, and D90 are particle size percentiles; they refer to particle sizes at which 10%, 50%, and 90% of the particles in the samples, respectively, are smaller than that size. In this context, the term Span does not refer to the surfactant, but instead is a common calculation generated by the software of all major particle size instruments including Malvern, Sympatec, Horiaba, and the Camsizer used here. It is used to represent a measure of how widely or narrowly distributed the particle size of a material is. The smaller the Span value, the narrower is the distribution of the material. In the context of this study, a narrower distribution is preferred, as it ensures that we can negate particle size as a factor which influences bulk characteristics.
  1. I have to live here, perhaps it should have been at the end, what is the goal here, the co-processed excipient, do we think about how it will affect the rate of fermentation in the final formulation and the dissolution of the medicinal substance. Whether our goal is perhaps the use of lipid excipients, which can have a positive effect on the dissolution rate, must be clarified. But the title is not clear. This should be clarified in the introduction and reinforced throughout the work.
  • The authors acknowledge that what the reviewer is discussing is an important goal of these types of formulations to improve the bioavailability and stability of active pharmaceutical ingredients. The manuscript introduction has been comprehensively re-written to clarify the scope and goal of this study has been to particle engineer prototype powders for their bulk powder behaviour whereas improvements to bioavailability and stabilisation are outside the scope of this study. The goal of this study was to determine the effect of ÊŸ-leucine surface modification on the flowability performance of different spray dried powder prototype excipient materials. This forms the basis for future studies investigating its impact on dissolution rates and bioavailability. We have also comprehensively revised the entire manuscript to accommodate this concern.
  1. 2. chapter little discussed
  • The authors acknowledge that this section was brief, and we have revised it accordingly to reflect the key points from the data. Its intention was to clarify minimal variation in particle size distributions between each of the spray-dried excipient/ÊŸ-leucine powders. This was essential for validating the study design by highlighting how particle size does not play a substantial role in influencing powder cohesion.
  1. I don't like the word formulation. This means the medicinal substance. Co-processed excipients.
  • The authors appreciate this comment and have revised use of the term formulation accordingly to clarify its reference to prototype powder compositions and relate with the term co-processed excipients. We note in general use, the term formulation is not exclusively limited to compositions including an active pharmaceutical and refers to the process of designing and developing a product or material by selecting and combining various components, ingredients, or substances in a specific way to achieve a desired set of properties or characteristics.
  1. Figure 7 units on both the x-axis and the y-axis. Pay attention to the discrete peaks that are observed, for example in the ACEG images around 10 degrees. The general shape is like this because of a lot of amorphous material, but if there is nothing crystalline, look at the more detailed diffractograms and potential, common, significant peaks.
  • The P-XRD section has been enhanced with additional references and discussion.
  1. Think again about the term wording
  • The authors have comprehensively revised the manuscript to resolve any wording, grammar, and abbreviation issues.
  1. See for yourself the non-uniformity in Figure 10.
  • The authors acknowledge the non-uniformity in the y-axis of Figure 10D, this is necessary to highlight the profound impact that the fibrous by-products in PVP K90/ ÊŸ-leucine powder has on overall flowability. It is also necessary to highlight the smaller, yet repeatable differences in the other carrier formulations.
  1. Check the results for some pictures, for example, 11f, there are illogical points, or explain them and support them with literature
  • The authors acknowledge the existence of these apparent anomalous points; they were repeatable in nature, and as yet we are still trying to understand and explain these anomalies. As a result, the figure has been removed pending further research and subsequent publication. A review of the literature did not provide a plausible explanation for their occurrence.
  1. Check references.
  • References were checked again to ensure compliance with the journal instructions.

We believe that our revised manuscript is now more comprehensive and will make a valuable contribution to this field. We would like to thank you for their guidance and recommendations, which have undoubtedly improved the quality of our manuscript.

Thank you again for your time and effort.

Reviewer 4 Report

The work presented by Danni Suhaidi and co-authors is interesting and well-presented. However, statistical analysis method is missing and must be added. Moreover, to improve the quality of this manuscripts some parts need to be considered as reported below.

Introduction: the state of art regarding the properties of polyvinylpyrrolidone could be improved adding information about its mucoadhesive property as it is a fundamental characteristic when novel delivery platforms were prepared for pharmaceutical purposed (please check: 10.3390/pharmaceutics11010035, 10.1016/j.ijbiomac.2012.07.009)

Line 141: please explain what TDS means and remove symbol not allowed (@), check in the entire manuscript.

Table 1: could the Authors change the sample formula code? In this form could be misleading for readers understand the type of excipient involved.

Line 150: please add the vacuum condition, detector and magnification parameters used during the SEM analyses.

Figure 1: samples based on maltodextrin containing leucin are not reported in the method sections, please add them.

Line 356: this part needs some clarification. First, the diffractogram performed on the sample without leucin obviously not display any peak as polymers have not crystallin rearrangement. This is an evidence, not a result. Second, the XRD analysis leucin alone is mandatory to understand if its crystallin form was maintained when entrapped into microparticles.

English and spell require minor checking and the statements are reported using a proper scientific language. 

Author Response

Thank you for your time and effort in reviewing this manuscript. We very much appreciate the input from your experience and insights in providing valuable comments and requested modifications, which have helped us improve the quality of our manuscript. We are pleased to inform you that we believe we have addressed your concerns and have revised the manuscript accordingly. Please find attached the revised manuscript, along with a detailed response to your comments. We have addressed your review point-by-point as follows:

  1. The work presented by Danni Suhaidi and co-authors is interesting and well-presented. However, statistical analysis method is missing and must be added. Moreover, to improve the quality of this manuscripts some parts need to be considered as reported below.

  • Thank you for providing this important feedback, a statistical analysis approach has been added into the method section and statistical analysis added into relevant sections of the results.

  1. Introduction: the state of art regarding the properties of polyvinylpyrrolidone could be improved adding information about its mucoadhesive property as it is a fundamental characteristic when novel delivery platforms were prepared for pharmaceutical purposed (please check: 10.3390/pharmaceutics11010035, 10.1016/j.ijbiomac.2012.07.009)

  • Thank you for providing this valuable advice, context describing the mucoadhesive properties of PVP added to the Introduction.

  1. Line 141: please explain what TDS means and remove symbol not allowed (@), check in the entire manuscript.
  • Oversight was corrected, and the proper term total dissolved solids was used in lieu of its abbreviation. The use of ‘@’ was removed from the entire manuscript.
  1. Table 1: could the Authors change the sample formula code? In this form could be misleading for readers understand the type of excipient involved.
  • Table 1 has been completely replaced with new terminology for the formulations to correct the confusion identified here.
  1. Line 150: please add the vacuum condition, detector and magnification parameters used during the SEM analyses.

  • The SEM formatting issue corrected, operating conditions, and magnification are now visible.

  1. Figure 1: samples based on maltodextrin containing leucin are not reported in the method sections, please add them.

  • Issues corrected in the new version of Table 1.

  1. Line 356: this part needs some clarification. First, the diffractogram performed on the sample without leucin obviously not display any peak as polymers have not crystallin rearrangement. This is an evidence, not a result. Second, the XRD analysis leucin alone is mandatory to understand if its crystallin form was maintained when entrapped into microparticles.

  • P-XRD diffractogram of pure crystalline ÊŸ-leucine added to supplement analysis. Section re-worked to better highlight past work which more comprehensively investigated P-XRD diffractograms from ÊŸ-leucine surface modification.

We believe that our revised manuscript is now more comprehensive and will make a valuable contribution to this field. We would like to thank you for their guidance and recommendations, which have undoubtedly improved the quality of our manuscript.

Thank you again for your time and effort.

Reviewer 5 Report

The manuscript presented by Suhaidi et al., showed interesting results regarding the surface modification of excipients used in amorphous solid dispersions. Such modifications were obtained using l-leucine during the spray drying process. In my opinion the manuscript is relevant. There is a need for studies optimizing manufacturing process parameters for ASD. In my opinion the manuscript could be accepted after two minor corrections.

Line 149. Table 1. Please reformulate. It is not clear the amount of leucine used for each excipient. For example, it seems that Maltodextrin was only evaluated with 0% leucine.

Line 315. Please include SEM of K100M as supplementary material.

Author Response

Thank you for your time and effort in reviewing this manuscript. We very much appreciate the input from your experience and insights in providing valuable comments and requested modifications, which have helped us improve the quality of our manuscript. We are pleased to inform you that we believe we have addressed your concerns and have revised the manuscript accordingly. Please find attached the revised manuscript, along with a detailed response to your comments. We have addressed your review point-by-point as follows:

  1. Line 149. Table 1. Please reformulate. It is not clear the amount of leucine used for each excipient. For example, it seems that Maltodextrin was only evaluated with 0% leucine.
  • Table 1 has been completely replaced with new terminology for the formulations to correct the confusion identified here.
  1. Line 315. Please include SEM of K100M as supplementary material.

  • HPMC K100M SEM images have been added as supplementary materials. They were originally not included because they displayed identical morphologies to the HPMC E5LV formulations.

We believe that our substantially revised manuscript is now more comprehensive and will make a valuable contribution to this field. We would like to thank you for their guidance and recommendations, which have undoubtedly improved the quality of our manuscript.

Thank you again for your time and effort.

Reviewer 6 Report

The manuscript entitled “Surface optimization of amorphous solid dispersion excipient powders” showed interesting data for preparation of solid dispersions of L-leucine and co-processing excipients, which can be applied as an excipient for the production of solid dosage forms. 

The study shown in this manuscript was well organized.  The authors varied the contents of all co-processing excipients at the same levels and compared the physicochemical properties of the obtained solid dispersions. However, there are some comments and questions for this manuscript as follows:

1. Why did the authors not add any hydrophobic drugs into the L-leucine and co-processing solid dispersion and then determine the solubility of such solid dispersions? In addition, the obtained results should be compared to the solubility of  the particular drugs (without addition of excipients)

2. The authors should include the results from physicochemical properties evaluation of physical mixtures of L-leucine and co-processing excipients containing the same contents of the compositions as the obtained solid dispersions in this manuscript.

3. The authors should investigate the interactions between molecules of the compositions in the obtained solid dispersions by using a suitable method, for example, the FT-IR spectroscopy, and then, show in the manuscript.

4. For the section 3.3. (Powder X-ray diffraction), the authors informed that the viscosity of HPMC affected the crystallization process of L-Leucine during spray drying. Therefore, the authors should show the viscosity of the HPMC system and the others.

5. According to the section 3.3. (Powder X-ray diffraction), Gum Arabic generally affects the viscosity of the solution for spray drying.  Why did the viscosity of Gum Arabic system not significantly affect the crystallization process of L-Leucine during spray drying like HPMC?

6. Please carefully check a word “physiochemical…” in the entire manuscript.  Should it be “physicochemical…”?

Author Response

Thank you for your time and effort in reviewing this manuscript. We very much appreciate the input from your experience and insights in providing valuable comments and requested modifications, which have helped us improve the quality of our manuscript. We are pleased to inform you that we believe we have addressed your concerns and have revised the manuscript accordingly. Please find attached the revised manuscript, along with a detailed response to your comments. We have addressed your review point-by-point as follows:

  1. Why did the authors not add any hydrophobic drugs into the L-leucine and co-processing solid dispersion and then determine the solubility of such solid dispersions? In addition, the obtained results should be compared to the solubility of the particular drugs (without addition of excipients)
  • The authors acknowledge that what the reviewer is discussing is an important goal of these types of formulations to improve the bioavailability and stability of active pharmaceutical ingredients. The manuscript introduction has been comprehensively re-written to clarify the scope and goal of this study has been to particle engineer prototype powders for their bulk powder behaviour, whereas this improvements to bioavailability is outside the scope of this current study. The goal of this study was to determine the effect of ÊŸ-leucine surface modification on the flowability performance of different spray dried powder prototype excipient materials. The identified aspects the reviewer makes forms the basis for future studies investigating its impact on dissolution rates and bioavailability. We have also comprehensively revised the entire manuscript to accommodate this concern.
  1. The authors should include the results from physicochemical properties evaluation of physical mixtures of L-leucine and co-processing excipients containing the same contents of the compositions as the obtained solid dispersions in this manuscript.
  • The reviewers raised an excellent point regarding the evaluation of the bulk flowability of the physical mixtures in the context of spray drying. It was concluded that physical mixtures do not provide an adequate benchmark comparison because of the mechanism of action behind ÊŸ-leucine surface modification. Therefore, to evaluate the effectiveness of ÊŸ-leucine co-spray drying, the authors decided that the best baseline for bulk characteristic comparison was spray-dried pure material. We have addressed this concern by clarification of comparison in the introduction.

  1. The authors should investigate the interactions between molecules of the compositions in the obtained solid dispersions by using a suitable method, for example, the FT-IR spectroscopy, and then, show in the manuscript.

  • The discussion section has been updated to include the importance of using spectroscopic techniques such as FT-IR and XPS to better understand the intermolecular interactions occurring within the formulated powder. This data is currently planned to be included in a follow-up publication focusing on the molecular basis of ÊŸ-leucine interactions in these compositions. The authors deemed this to be out of the scope of this study, and the detail of such spectroscopic techniques will result in the manuscript being excessively long and peripheral given the manuscript primarily focuses on the impact of ÊŸ-leucine on bulk powder characteristics.

  1. For the section 3.3. (Powder X-ray diffraction), the authors informed that the viscosity of HPMC affected the crystallization process of L-Leucine during spray drying. Therefore, the authors should show the viscosity of the HPMC system and the others.

  • We have recognised the reviewer's request for viscosity information to support their claims. Unfortunately, measuring viscosity would only be applicable to formulations before spray drying, as the experimental design purposely utilised excess water to minimise the impact of viscosity. The process described, however, refers to what occurs during the spray-drying process itself, in which the droplet viscosity is substantially more complicated to quantify. We intend to investigate this aspect more in subsequent publications.

  1. According to the section 3.3. (Powder X-ray diffraction), Gum Arabic generally affects the viscosity of the solution for spray drying. Why did the viscosity of Gum Arabic system not significantly affect the crystallization process of L-Leucine during spray drying like HPMC?
  • The reviewer made a very valid point regarding the potential impact of gum arabic on the solution viscosity during spray drying. This goes to the core purpose of our study which highlights the gap in understanding of phenomena such as viscosity impact on spray dried powder cohesion. The authors did not observe such an impact during feedstock preparation which we attribute to the use of excess water which minimises viscosity effects. Although the literature suggests that gum arabic should have a higher viscosity than HPMC E5LV under similar conditions, it did not display the same level of disruptive behaviour as HPMCs in this study. The authors proposed that several factors could contribute to this difference, such as variations in temperature dependence and gelation behaviour between the two materials or differences in the functional groups present in their molecular structures, potentially affecting their interactions with l-leucine. However, the exact mechanism behind this phenomenon remains uncertain, and further exploration through spectroscopic analysis is necessary to enhance our understanding of l-leucine surface modification. We have clarified in the discussion the aim to focus on these aspects in future studies.
  1. Please carefully check a word “physiochemical…” in the entire manuscript. Should it be “physicochemical…”?

  • Term ‘physiochemical’ replaced with ‘physico-chemical’.

We believe that our revised manuscript is now more comprehensive and will make a valuable contribution to this field. We would like to thank you for their guidance and recommendations, which have undoubtedly improved the quality of our manuscript.

Thank you again for your time and effort.

Round 2

Reviewer 2 Report

I accept that the current version of the manuscript is much better written and scientifically more acceptable than “version 1” of the manuscript.

Unfortunately, there are still countless mistakes in this manuscript; both scientific and grammatical!

SORRY, ABOUT THIS!

Examples, include-but are not limited to - the following.

Surely it would take the authors less than an, about, 20 min to correct all of them instead of having to rely on the “copy-editor” of the journal?

1. Why write: “Currently, it was co-formulated in several………” (line 100); rather than: “Currently, it is co-formulated in several………”

2. Why are “Gum Arabic” (line 142), “Trehalose Powder” (line144) and Hydroxypropyl (line 148) capitalized?

3. SEM, is introduced as an abbreviation (line 136); why then reintroduce this abbreviation (line 168)?

4. There should be a space between values and units for 9kPa and 7kPa (lines 207/209, respectively); similarly, 15kPa (line 252).

5. Why not use the previously introduced and accepted abbreviation for L-Leucine (L-Leu; line 261), throughout the manuscript?

6. Why capitalize ……“Time-of-Flight Secondary Ion Mass Spectrometry”…… (line 935)?

7.  “……. inverse gas chromatography.” The foregoing is not a spectroscopic technique (line 935)!

Could still be improved.

Author Response

We thank the reviewer for the attention to detail and have continued to refine the consistency, terminology, grammatical and writing quality of the manuscript.

Reviewer 3 Report

The quality of work has improved significantly. I believe that the main function of the new excipient should have been investigated, which is the improvement of the dissolution rate. In this way, I would still change the title with an emphasis on improving the characteristics and functionality of excipients... and emphasize that throughout the paper. This is the abstract, the second sentence, the essence was not shown to me anywhere.

The quality of work has improved significantly. I believe that the main function of the new excipient should have been investigated, which is the improvement of the dissolution rate. In this way, I would still change the title with an emphasis on improving the characteristics and functionality of excipients... and emphasize that throughout the paper. This is the abstract, the second sentence, the essence was not shown to me anywhere.

Author Response

We really appreciate and agree the view of the reviewer that the impact of dissolution rate is appropriate in the wider context of work of this nature, however we disagree that it is appropriate to include in the current paper given the current focus is purely on the highly complex nature of bulk powder cohesion, its optimisation and characterisation. We clarified this in the paper through altering the title and elements of the introduction to reinforce the focus on the paper is to address challenges of bulk powder behaviour. We have also clarified that amorphous solid dispersions are not only used in improving dissolution of poorly soluble drugs, they are also used widely in stabilisation of biomacromolecules and hence not only would dissolution studies be appropriate in subsequent work, but also evaluation of chemical stability with relevant macromolecules.

Reviewer 4 Report

The new version of this manuscript is greatly improved. Introduction, abstract and method sections are now well written. I Appreciate that Authors have considered seriously the comments and modified the work taking them into account. 

Typos still need to be fixed and the entire English language should be checked.

Author Response

(The authors gave the same response as above.)

Reviewer 6 Report

The authors insisted to present their works in the present form. I thus accept  their works to publish in the present form.

Author Response

We thank and appreciate the reviewers attention and effort during this peer-review process.